# Projecting genetic associations through gene expression patterns highlights disease etiology and drug mechanisms

Milton Pividori [1,2], Sumei Lu[3], Binglan Li [4], Chun Su[3], Matthew E. Johnson[3], Wei-Qi Wei [5], Qiping Feng [5], Bahram Namjou [6], Krzysztof Kiryluk[7], Iftikhar J. Kullo [8], Yuan Luo [9], Blair D. Sullivan [10], Benjamin F. Voight [1,11,12], Carsten Skarke [12], Marylyn D. Ritchie [1], Struan F. A. Grant [1,3,13,14,15], eMERGE Consortium* & Casey S. Greene [2,16] ✉

Genes act in concert with each other in specific contexts to perform their functions. Determining how these genes influence complex traits requires a mechanistic understanding of expression regulation across different conditions. It has been shown that this insight is critical for developing new therapies. Transcriptome-wide association studies have helped uncover the role of individual genes in disease-relevant mechanisms. However, modern models of the architecture of complex traits predict that gene-gene interactions play a crucial role in disease origin and progression. Here we introduce PhenoPLIER, a computational approach that maps gene-trait associations and pharmacological perturbation data into a common latent representation for a joint analysis. This representation is based on modules of genes with similar expression patterns across the same conditions. We observe that diseases are significantly associated with gene modules expressed in relevant cell types, and our approach is accurate in predicting known drug-disease pairs and inferring mechanisms of action. Furthermore, using a CRISPR screen to analyze lipid regulation, we find that functionally important players lack associations but are prioritized in trait-associated modules by PhenoPLIER. By incorporating groups of co-expressed genes, PhenoPLIER can contextualize genetic associations and reveal potential targets missed by single-gene strategies.

Genes work together in context-specific networks to carry out different functions[1,2]. Variations in these genes can change their functional role and, at a higher level, affect disease-relevant biological processes[3]. In this context, determining how genes influence complex traits requires mechanistically understanding expression regulation across different cell types[4–6], which in turn should lead to improved treatments[7,8]. Previous studies have described different regulatory DNA elements[5,9–12] including genetic effects on gene expression across different tissues[4]. Integrating functional genomics data and GWAS data[13–15] has improved the identification of these transcriptional mechanisms that, when dysregulated, commonly result in tissue- and cell lineage-specific pathology[16–18].

Given the availability of gene expression data across several tissues[4,19–21], an effective approach to identify these biological

A full list of affiliations appears at the end of the paper. *A list of authors and their affiliations appears at the end of the paper.
✉e-mail: casey.s.greene@cuanschutz.edu

processes is the transcription-wide association study (TWAS), which integrates expression quantitative trait loci (eQTLs) data to provide a mechanistic interpretation for GWAS findings. TWAS relies on testing whether perturbations in gene regulatory mechanisms mediate the association between genetic variants and human diseases[22–25], and these approaches have been highly successful not only in understanding disease etiology at the transcriptome level[26–28] but also in disease-risk prediction (polygenic scores)[29] and drug-repurposing[30] tasks. However, TWAS works at the individual gene level, which does not capture more complex interactions at the network level.

These gene-gene interactions play a crucial role in current theories of the architecture of complex traits, such as the omnigenic model[31], which suggests that methods need to incorporate this complexity to disentangle disease-relevant mechanisms. Widespread gene pleiotropy, for instance, reveals the highly interconnected nature of transcriptional networks[32,33], where potentially all genes expressed in disease-relevant cell types have a non-zero effect on the trait[31,34]. One way to learn these gene-gene interactions is using the concept of gene module: a group of genes with similar expression profiles across different conditions[2,35,36]. In this context, several unsupervised approaches have been proposed to infer these gene-gene connections by extracting gene modules from co-expression patterns[37–39]. Matrix factorization techniques like independent or principal component analysis (ICA/PCA) have shown superior performance in this task[40] since they capture local expression effects from a subset of samples and can handle modules overlap effectively. Therefore, integrating genetic studies with gene modules extracted using unsupervised learning could further improve our understanding of disease origin[36] and progression[41].

Here we propose PhenoPLIER, an omnigenic approach that provides a gene module perspective to genetic studies. The flexibility of our method allows integration of different data modalities into the same representation for a joint analysis. We show that this module perspective can infer how groups of functionally related genes influence complex traits, detect shared and distinct transcriptomic properties among traits, and predict how pharmacological perturbations affect genes' activity to exert their effects. PhenoPLIER maps gene-trait associations and drug-induced transcriptional responses into a common latent representation. For this, we integrate thousands of gene-trait associations (using TWAS from PhenomeXcan[42]) and transcriptional profiles of drugs (from LINCS L1000[43]) into a low-dimensional space learned from public gene expression data on tens of thousands of RNA-seq samples (recount2[19,44]). We use a latent representation defined by a matrix factorization approach[44,45] that extracts gene modules with certain sparsity constraints and preferences for those that align with prior knowledge (pathways). When mapping gene-trait associations to this reduced expression space, we observe that diseases are significantly associated with gene modules expressed in relevant cell types: such as hypothyroidism with T cells, corneal endothelial cells with keratometry measurements, hematological assays on specific blood cell types, plasma lipids with adipose tissue, and neuropsychiatric disorders with different brain cell types. Moreover, since PhenoPLIER can use models derived from large and heterogeneous RNA-seq datasets, we can also identify modules associated with cell types under specific stimuli or disease states. We observe that significant module-trait associations in PhenomeXcan (our discovery cohort) replicated in the Electronic Medical Records and Genomics (eMERGE) network phase III[27,46] (our replication cohort). Furthermore, we perform a CRISPR screen to analyze lipid regulation in HepG2 cells. We observe more robust trait associations with modules than with individual genes, even when single genes known to be involved in lipid metabolism did not reach genome-wide significance. Compared to a single-gene approach, our module-based method also better predicts FDA-approved drug-disease links by capturing tissue-specific pathophysiological mechanisms linked with the mechanism of action of

drugs (e.g., niacin with cardiovascular traits via a known immune mechanism). This improved drug-disease prediction suggests that modules may provide a better means to examine drug-disease relationships than individual genes. Finally, exploring the phenotype-module space reveals stable trait clusters associated with relevant tissues, including a complex branch involving lipids with cardiovascular, autoimmune, and neuropsychiatric disorders. In summary, instead of considering single genes associated with different complex traits, PhenoPLIER incorporates groups of genes that act together to carry out different functions in specific cell types. This approach improves robustness in detecting and interpreting genetic associations, and here we show how it can prioritize alternative and potentially more promising candidate targets even when known single-gene associations are not detected. The approach represents a conceptual shift in the interpretation of genetic studies. It has the potential to extract mechanistic insight from statistical associations to enhance the understanding of complex diseases and their therapeutic modalities.

## Results

### PhenoPLIER: an integration framework based on gene co-expression patterns

PhenoPLIER is a flexible computational framework that combines gene-trait and gene-drug associations with gene modules expressed in specific contexts (Fig. 1a). The approach uses a latent representation (with latent variables or LVs representing gene modules) derived from a large gene expression compendium (Fig. 1b, top) to integrate TWAS with drug-induced transcriptional responses (Fig. 1b, middle) for a joint analysis. The approach consists of three main components (Fig. 1b, bottom, see Methods): (1) an LV-based regression model to compute an association between an LV and a trait, (2) a clustering framework to learn groups of traits with shared transcriptomic properties, and (3) an LV-based drug-repurposing approach that links diseases to potential treatments. We performed extensive simulations for our regression model (Supplementary Note 1) and clustering framework (Supplementary Note 2) to ensure proper calibration and expected results under a model of no association.

We used TWAS results from PhenomeXcan[42] and the eMERGE network[27] as discovery and replication cohorts, respectively (Methods). PhenomeXcan provides gene-trait associations for 4091 different diseases and traits from the UK Biobank[47] and other studies, whereas the analyses on eMERGE were performed across 309 phecodes. TWAS results were derived using two statistical methods (see Methods): (1) Summary-MultiXcan (S-MultiXcan) associations were used for the regression and clustering components, and (2) Summary-PrediXcan (S-PrediXcan) associations were used for the drug-repurposing component. In addition, we also used colocalization results, which provide a probability of overlap between the GWAS and eQTL signals. For the drug-repurposing approach, we used transcriptional responses to small molecule perturbations from LINCS L1000[43] comprising 1170 compounds.

The latent gene expression representation was obtained from the MultiPLIER models[44], which were derived by applying a matrix factorization method (the pathway-level information extractor or PLIER[45]) to recount2[19]—a uniformly-curated collection of transcript-level gene expression quantified by RNA-seq in a large, diverse set of samples collected across a range of disease states, cell types differentiation stages, and various stimuli (see Methods). The MultiPLIER models extracted 987 LVs by optimizing data reconstruction but also the alignment of LVs with prior knowledge/pathways.

Each LV or gene module represents a group of weighted genes expressed together in the same tissues and cell types as a functional unit. Since LVs might represent a functional set of genes regulated by the same transcriptional program[48,49], we conjecture that the projection of TWAS and pharmacologic perturbations data into this latent space could provide a better mechanistic understanding. For this

projection of different data modalities into the same space, PhenoPLIER converts gene associations to an LV score: all genes' standardized effect sizes for a trait (from TWAS) or differential expression values for a drug (from pharmacologic perturbation data) are multiplied by the LV genes' weights and summed, producing a single value. Instead of looking at individual genes, this process links different traits and drugs to functionally related groups of genes or LVs. PhenoPLIER uses LVs annotations about the specific conditions where the group of genes is expressed, such as cell types and tissues, even at specific developmental stages, disease stages or under distinct stimuli. Although this is not strictly necessary for PhenoPLIER to work, these annotations can dramatically improve the interpretability of results. MultiPLIER's models provide this information by linking LVs to samples, which may be annotated for experimental conditions (represented by matrix **B** at the top of Fig. 1b) in which genes in an LV are expressed. An example of this is shown in Fig. 1c. In the original MultiPLIER study, the authors reported that one of the latent variables, identified as LV603, was associated with a known neutrophil pathway and highly correlated with neutrophil count estimates from whole blood RNA-seq profiles[50]. We analyzed LV603 using PhenoPLIER and found that 1) neutrophil counts and other white blood cell traits were ranked among the top 10 traits out of 4091 (Fig. 1c, bottom), and basophils count and percentage were significantly associated with this LV when using our regression method (Supplementary Table 4), and 2) LV603's genes were expressed in highly relevant cell types

(Fig. 1c, top). These initial results suggested that groups of functionally related and co-expressed genes tend to correspond to groups of trait-associated genes, and the approach can link transcriptional mechanisms from large and diverse dataset collections to complex traits.

Therefore, PhenoPLIER allows the user to address specific questions, namely: do disease-associated genes belong to modules expressed in specific tissues and cell types? Are these cell type-specific modules associated with different diseases, thus potentially representing a "network pleiotropy" example from an omnigenic point of view[31]? Is there a subset of module's genes that is closer to the definition of "core" genes (i.e., directly affecting the trait with no mediated regulation of other genes[34]) and thus represents alternative and potentially better candidate targets? Are drugs perturbing these transcriptional mechanisms, and can they suggest potential mechanisms of action?

## LVs link genes that alter lipid accumulation with relevant traits and tissues

Our first experiment attempted to answer whether genes in a disease-relevant LV could represent potential therapeutic targets. For this, the first step was to obtain a set of genes strongly associated with a phenotype of interest. Therefore, we performed a fluorescence-based CRISPR-Cas9 in the HepG2 cell line and identified 462 genes associated with lipid regulation (Methods). From these, we selected two high-confidence gene sets that either caused a decrease or increase of lipids:

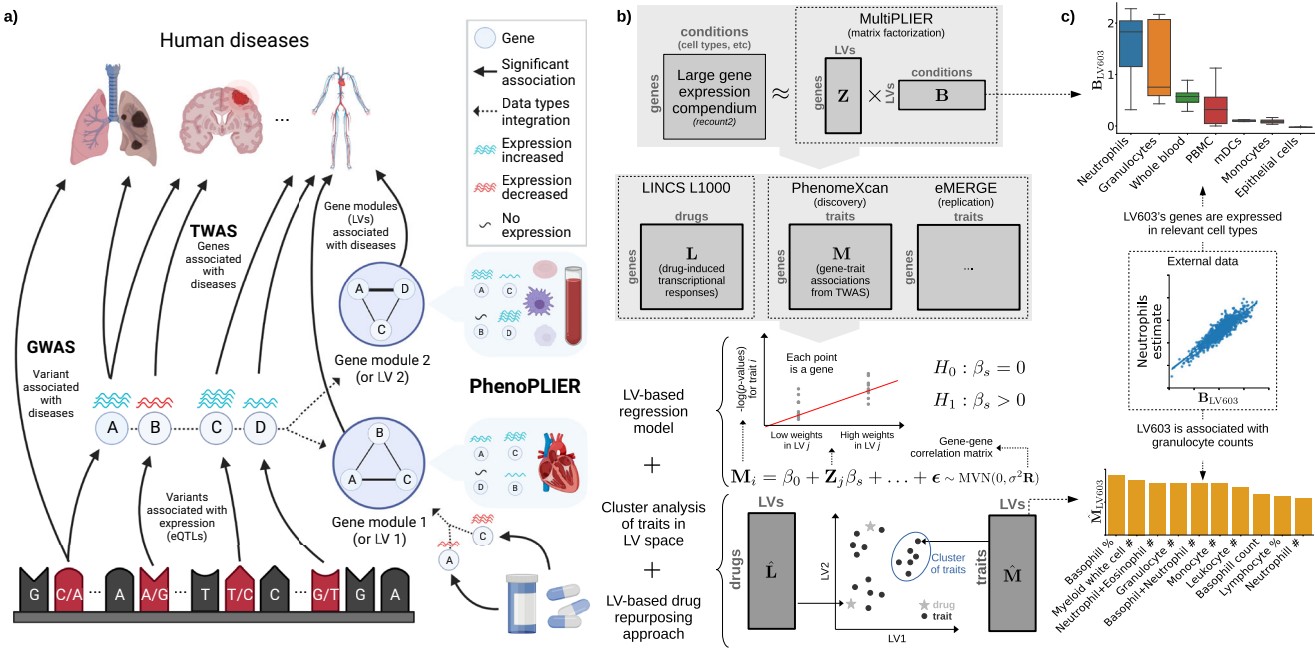

**Fig. 1 | Schematic of the PhenoPLIER framework. a** High-level schematic of PhenoPLIER (a gene module-based method) in the context of TWAS (single-gene) and GWAS (single-variant). In GWAS, we identify variants associated with traits. In TWAS, first, we identify variants that are associated with gene expression levels (eQTLs); then, prediction models based on eQTLs are used to impute gene expression, which is used to compute gene-trait associations. Resources such as LINCS L1000 provide information about how a drug perturbs gene expression; at the bottom-right corner, we show how a drug downregulates two genes (A and C). In PhenoPLIER, these data types are integrated using groups of genes co-expressed across one or more conditions (such as cell types) that we call gene modules or latent variables/LVs. Created with BioRender.com. **b** The integration process in PhenoPLIER uses low-dimensional representations (matrices **Z** and **B**) learned from large gene expression datasets (top). We used gene-drug information **L** from LINCS L1000 and gene-trait associations **M** from TWAS: PhenomeXcan was used as the discovery cohort, and eMERGE as replication (middle). PhenoPLIER provides three computational components (bottom): 1) an LV-based regression model that associates an LV $j$ ($\mathbf{Z}_j$) with a trait $i$ ($\mathbf{M}_i$), 2) a clustering framework that learns groups of traits from TWAS associations projected into the LV space ($\hat{\mathbf{M}}$), and 3) an LV-based drug-repurposing approach that uses the projection of TWAS ($\hat{\mathbf{M}}$) and LINCS L1000 ($\hat{\mathbf{L}}$) into the LV space. **c** Genes that are part of LV603, termed as a neutrophil signature[44], were expressed in relevant cell types (top), with 53 independent samples expressed in Neutrophils, 59 in Granulocytes, and 20 in Whole blood, 56 in PBMC, 8 in mDCs, 29 in Monocytes, and 5 in Epithelial cells (the boxplot shows the 25th, 50th and 75th percentiles while the whiskers extend to the minimum/maximum values). LV603 was associated in PhenoPLIER with neutrophil counts and other white blood cells (bottom, showing the top 10 traits for LV603 after projecting gene-trait associations in PhenomeXcan). eQTLs expression quantitative trait loci, MVN multivariate normal distribution, PBMC peripheral blood mononuclear cells, mDCs myeloid dendritic cells.

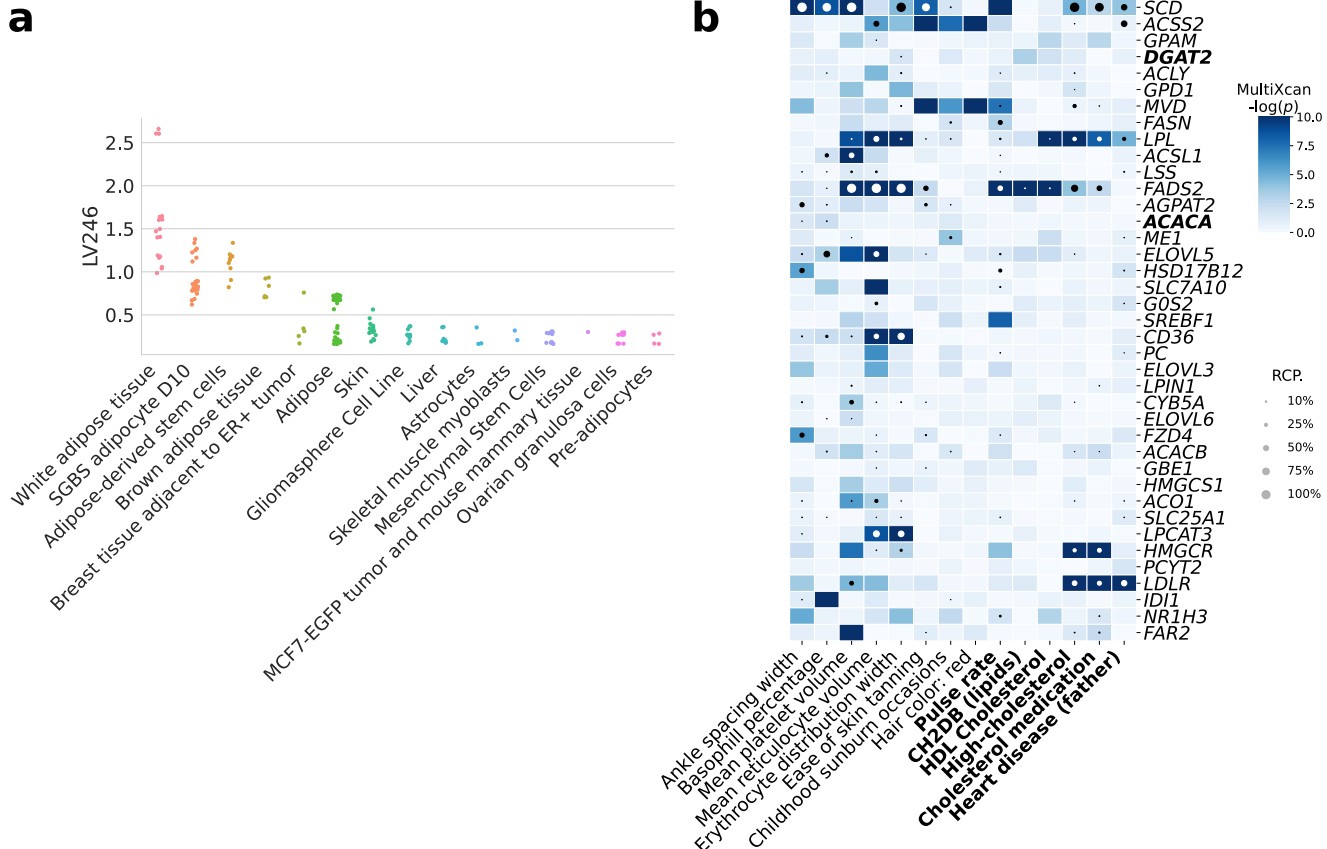

**Fig. 2 | Tissues and traits associated with a gene module related to lipid metabolism (LV246). a** Top cell types/tissues in which LV246's genes are expressed. Values in the *y* axis come from matrix **B** in the MultiPLIER models (Fig. 1b, see Methods). In the *x* axis, cell types/tissues are sorted by the maximum sample value. **b** Gene-trait associations (unadjusted *p*-values from S-MultiXcan[24]; threshold at -log(*p*) = 10) and colocalization probability (fastENLOC) for the top traits in LV246.

The top 40 genes in LV246 are shown, sorted by their LV weight (matrix **Z**), from largest (the top gene *SCD*) to smallest (*FAR2*); *DGAT2* and *ACACA*, in boldface, are two of the six high-confidence genes in the lipids-increasing gene set from the CRISPR screen. Cardiovascular-related traits are in boldface. SGBS Simpson Golabi Behmel Syndrome, CH2DB CH2 groups to double bonds ratio, HDL high-density lipoprotein, RCP locus regional colocalization probability.

a lipids-decreasing gene set with eight genes: *BLCAP, FBXW7, INSIG2, PCYT2, PTEN, SOX9, TCF7L2, UBE2J2*; and a lipids-increasing gene set with six genes: *ACACA, DGAT2, HILPDA, MBTPS1, SCAP, SRPR* (Supplementary Data 2).

Next, we analyzed all 987 LVs using Fast Gene Set Enrichment Analysis (FGSEA)[51], and found 15 LVs nominally enriched (unadjusted *P* < 0.01) with these lipid-altering gene sets (Supplementary Tables 1 and 2). Among those with reliable sample metadata, LV246, the top LV associated with the lipids-increasing gene set, contained genes mainly co-expressed in adipose tissue (Fig. 2a), which plays a key role in coordinating and regulating lipid metabolism. Using our regression framework across all traits in PhenomeXcan, we found that gene weights for this LV were predictive of gene associations for plasma lipids, high cholesterol, and Alzheimer's disease (Supplementary Table 7, FDR < 1e-23). These lipids-related associations also replicated across the 309 traits in eMERGE (Supplementary Table 8), where LV246 was significantly associated with hypercholesterolemia (phecode: 272.11, FDR < 4e-9), hyperlipidemia (phecode: 272.1, FDR < 4e-7) and disorders of lipoid metabolism (phecode: 272, FDR < 4e-7).

Two high-confidence genes from our CRISPR screening, *DGAT2,* and *ACACA*, are responsible for encoding enzymes for triglycerides and fatty acid synthesis and were among the highest-weighted genes of LV246 (Fig. 2b, in boldface). However, in contrast to other members of LV246, *DGAT2,* and *ACACA* were not associated nor colocalized with any of the cardiovascular-related traits and thus would not have been prioritized by TWAS alone; instead, other members of LV246, such as *SCD, LPL, FADS2, HMGCR,* and *LDLR*, were significantly associated and

colocalized with lipid-related traits. This lack of association of two high-confidence genes from our CRISPR screen might be explained from an omnigenic point of view[34]. Assuming that the TWAS models for *DGAT2* and *ACACA* capture all common *cis*-eQTLs (the only genetic component of gene expression that TWAS can capture) and there are no rare *cis*-eQTLs, these two genes might represent "core" genes (i.e., they directly affect the trait with no mediated regulation of other genes), and many of the rest in the LV are "peripheral" genes that *trans*-regulate them.

## LVs predict drug-disease pairs better than single genes
We next determined how substituting LVs for individual genes predicted known treatment-disease relationships. For this, we used the transcriptional responses to small molecule perturbations profiled in LINCS L1000[43], which were further processed and mapped to Drug-Bank IDs[52–54]. Based on an established drug-repurposing strategy that matches reversed transcriptome patterns between genes and drug-induced perturbations[55,56], we adopted a previously described framework that uses imputed transcriptomes from TWAS to prioritize drug candidates[30]. For this, we computed a drug-disease score by calculating the negative dot product between the *z*-scores for a disease (from TWAS) and the *z* scores for a drug (from LINCS) across sets of genes of different sizes (see Methods). Therefore, a large score for a drug-disease pair indicated that higher (lower) predicted expression values of disease-associated genes are down (up)-regulated by the drug, thus predicting a potential treatment. Similarly, for the LV-based approach, we estimated how pharmacological perturbations affected the gene

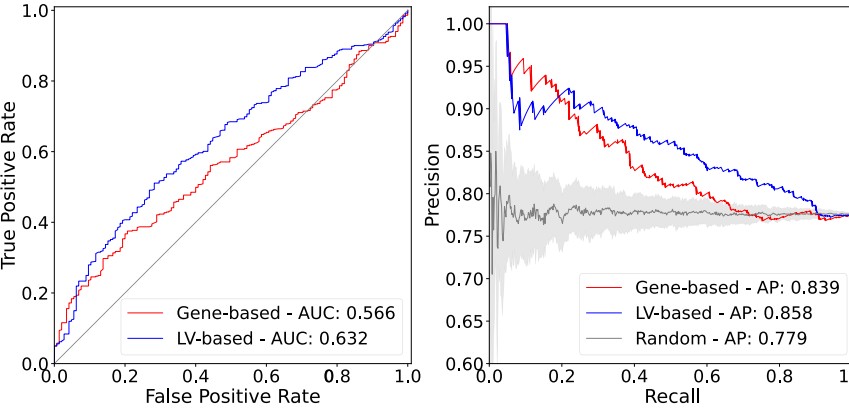

**Fig. 3 | Drug-disease prediction performance for gene-based and LV-based approaches.** The receiver operating characteristic (ROC) (left) and the precision-recall curves (right) for a gene-based and LV-based approach. "Random" refers to the average precision of a hundred classifiers with randomly permuted scores, where the error band represents the 95% confidence interval. AUC area under the curve; AP average precision.

module activity by projecting expression profiles of drugs into our latent representation (Fig. 1b). We used a manually curated gold standard set of drug-disease medical indications[53,57] for 322 drugs across 53 diseases to evaluate the prediction performance.

It is important to note that the gene-trait associations and drug-induced expression profiles projected into the latent space represent a compressed version of the entire set of results. Despite this information loss, the LV-based method outperformed the gene-based one with an area under the curve of 0.632 and an average precision of 0.858 (Fig. 3). The prediction results suggested that this low-dimensional space captures biologically meaningful patterns that can link pathophysiological processes with the mechanism of action of drugs.

We examined a specific drug-disease pair to determine whether the LVs driving the prediction were biologically plausible. Nicotinic acid (niacin) is a B vitamin widely used clinically to treat lipid disorders, although there is controversy on its clinical utility in preventing cardiovascular disease[58–60]. Niacin exerts its effects on multiple tissues, although its mechanisms are not well understood[61–64]. This compound can increase high-density lipoprotein (HDL) by inhibiting an HDL catabolism receptor in the liver. Niacin also inhibits diacylglycerol acyltransferase−2 (DGAT2), which decreases the production of low-density lipoproteins (LDL) either by modulating triglyceride synthesis in hepatocytes or by inhibiting adipocyte triglyceride lipolysis[61]. Niacin was one of the drugs in the gold standard set indicated for atherosclerosis (AT) and coronary artery disease (CAD). We observed that this compound was predicted by the gene-based and LV-based approach as a medical indication for coronary artery disease (CAD), with scores above the mean (0.51 and 0.96, respectively). For AT, the LV-based approach predicted niacin as a therapeutic drug with a score of 0.52, whereas the gene-based method assigned a negative score of -0.01 (below the mean). Since LVs represent interpretable features associated with specific cell types, we analyzed which LVs positively contributed to these predictions (i.e., with an opposite direction between niacin and the disease). Notably, LV246 (Fig. 2), expressed in adipose tissue and liver and associated with plasma lipids and high cholesterol (Supplementary Table 7), was the 16th most important module in the prediction of niacin as a therapeutic drug for AT. Besides the gold standard set, LV246 was among the top modules for other cardiovascular diseases, such as ischemic heart disease (wide definition, 15th module) and high cholesterol (7th module).

The analysis of other top niacin-contributing LVs across different cardiovascular diseases revealed additional mechanisms of action. For example, *GPR109A/HCAR2* encodes a G protein-coupled high-affinity niacin receptor in adipocytes and immune cells, including monocytes, macrophages, neutrophils, and dendritic cells[65,66]. It was initially

thought that the antiatherogenic effects of niacin were solely due to the inhibition of lipolysis in adipose tissue. However, it has been shown that nicotinic acid can reduce atherosclerosis progression independently of its antidyslipidemic activity by activating *GPR109A* in immune cells[67], thus boosting anti-inflammatory processes[68]. In addition, flushing, a common adverse effect of niacin, is also produced by the activation of GPR109A in Langerhans cells (macrophages of the skin). This alternative mechanism for niacin could have been hypothesized by examining the cell types where the top-contributing modules are expressed: for instance, LV116 and LV931 (Fig. 4, Supplementary Figure 15, and Supplementary Tables 9 and 10) were the top two modules for AT, with a strong signature in monocytes, macrophages, neutrophils, dendritic cells, among others. In Fig. 4, it can be seen that LV116's genes are expressed as an immune response when these cell types are under different stimuli, such as diarrhea caused by different pathogens[69], samples from multiple sclerosis or systemic lupus erythematosus[70,71], or infected with different viruses (such as herpes simplex[72], West Nile virus[73], *Salmonella typhimurium*[74], among others). These three LVs (LV246, LV116, and LV931) were among the top 20 modules contributing to the niacin prediction across different cardiovascular traits (Table 1).

Beyond cardiovascular traits, there are other potentially interesting LVs that could extend our understanding of the mechanisms of niacin. For example, LV66, one of the top LVs affected by niacin (Supplementary Figure 16), was mainly expressed in ovarian granulosa cells. This compound has been very recently considered a potential therapeutic for ovarian diseases[75,76], as it was found to promote follicle growth and inhibit granulosa cell apoptosis in animal models.

## LVs reveal trait clusters with shared transcriptomic properties

We used the projection of gene-trait associations into the latent space to find groups of clusters linked by the same transcriptional processes. Since individual clustering algorithms have different biases (i.e., assumptions about the data structure), we designed a consensus clustering framework that combines solutions or partitions of traits generated by different methods (Methods). Consensus or ensemble approaches have been recommended to avoid several pitfalls when performing cluster analysis on biological data[77]. Since diversity in the ensemble is crucial for these methods, we generated different data versions which were processed using different methods with varying sets of parameters (Fig. 5a). Then, a consensus function combines the ensemble into a consolidated solution, which has been shown to outperform any individual member of the ensemble[78,79]. Our clustering pipeline generated 15 final consensus clustering solutions (Supplementary Figure 13). The number of clusters of these partitions

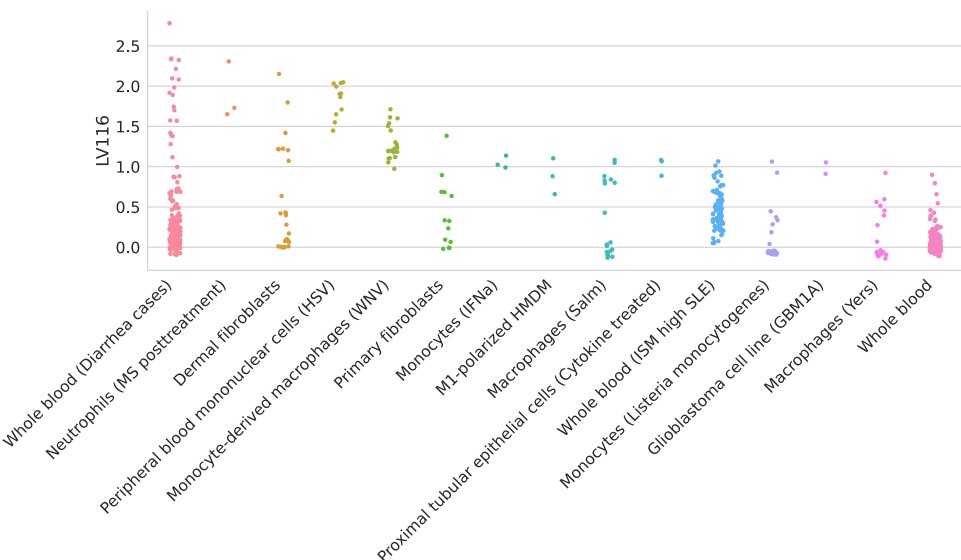

**Fig. 4 | Top cell types/tissues where LV116's genes are expressed.** Values in the *y* axis come from matrix **B** in the MultiPLIER models (Fig. 1b). In the *x* axis, cell types/ tissues are sorted by the maximum sample value. The figure shows a clear immune response with cell types under different stimuli. MS multiple sclerosis, HSV-treated with herpes simplex virus, WNV infected with West Nile virus, IFNa treated with interferon-alpha, HMDM human peripheral blood mononuclear cell-derived macrophages, Salm infected with *Salmonella typhimurium*, Yers infected with *Yersinia pseudotuberculosis*, ISM Interferon Signature Metric, SLE Systemic lupus erythematosus.

**Table 1 | LVs among the top 20 contributors to the prediction of niacin for five cardiovascular diseases and related traits**

| LV | Cell type | Disease |
|---|---|---|
| LV116 | Immune cells, skin | Atherosclerosis (ICD10 I70) |
| | | Chronic ischemic heart disease (ICD10 I25) |
| | | Heart attack, angina, stroke or hypertension |
| | | Ischemic heart disease (wide definition) |
| LV931 | Immune cells | Atherosclerosis (ICD10 I70) |
| | | Heart attack, angina, stroke or hypertension |
| | | Ischemic heart disease (wide definition) |
| LV246 | Adipose tissue, liver | Atherosclerosis (ICD10 I70) |
| | | High cholesterol (self-reported) |
| | | Ischemic heart disease (wide definition) |

"Heart attack, angina, stroke or hypertension" refers to the UK Biobank data-field 6150. GWAS sample size: Atherosclerosis (361,194 in total and 566 cases), Chronic ischemic heart disease (361,194 in total and 12,769 cases), Heart attack, angina, stroke or hypertension (360,420 in total and 253,565 cases), Ischemic heart disease/wide definition (361,194 in total and 20,857 cases), High cholesterol/ self-reported (361,141 in total and 43,957 cases).

(between 5 and 29) was learned from the data by selecting the partitions with the largest agreement with the ensemble[78]. Instead of selecting one of these final solutions with a specific number of clusters, we used a clustering tree[80] (Fig. 6) to examine stable groups of traits across multiple resolutions. To understand which latent variables differentiated the group of traits, we trained a decision tree classifier on the input data $\hat{\mathbf{M}}$ using the clusters found as labels (Fig. 5b, see Methods).

We found that phenotypes were grouped into five clear branches, defined by their first node at the top of Fig. 6: (0) a "large" branch that includes most of the traits subdivided only starting at $k = 16$ (with asthma, subjective well-being traits, and nutrient intake clusters), (1) heel bone-densitometry measurements, (2) hematological assays on red blood cells, (3) physical measures, including spirometry and body impedance, and anthropometric traits with fat-free and fat mass measures in separate sub-branches, and (4) a "complex" branch including keratometry measurements, assays on white blood cells and platelets, skin and hair color traits, autoimmune disorders, and cardiovascular diseases (which also included other cardiovascular-related

traits such as hand-grip strength[81], and environmental/behavioral factors such as physical activity and diet) (see Supplementary Data 3–7 for all clustering results). Within these branches, results were relatively stable, with the same traits often clustered together across different resolutions. Arrows between clusters show traits moving from one group to another, and this mainly happens between clusters within the "complex" branch (4) and between clusters from the "large" branch (0) to the "complex" branch. This behavior is expected since complex diseases are usually associated with shared genetic and environmental factors and are thus hard to categorize into a single cluster.

Next, we analyzed which LVs were driving these clusters of traits. For this, we trained decision tree classifiers on the input data using each cluster at $k = 29$ (bottom of Fig. 6) as labels (see Methods). This procedure yielded the top LVs that were most discriminative for each cluster. Several of these LVs were well-aligned to existing pathways (Fig. 7), whereas others were not aligned to prior knowledge but still expressed in relevant tissues (Supplementary Figure 14). In Fig. 7, it can be seen that some LVs were highly specific to certain traits, while others were associated with a wide range of different phenotypes, thus

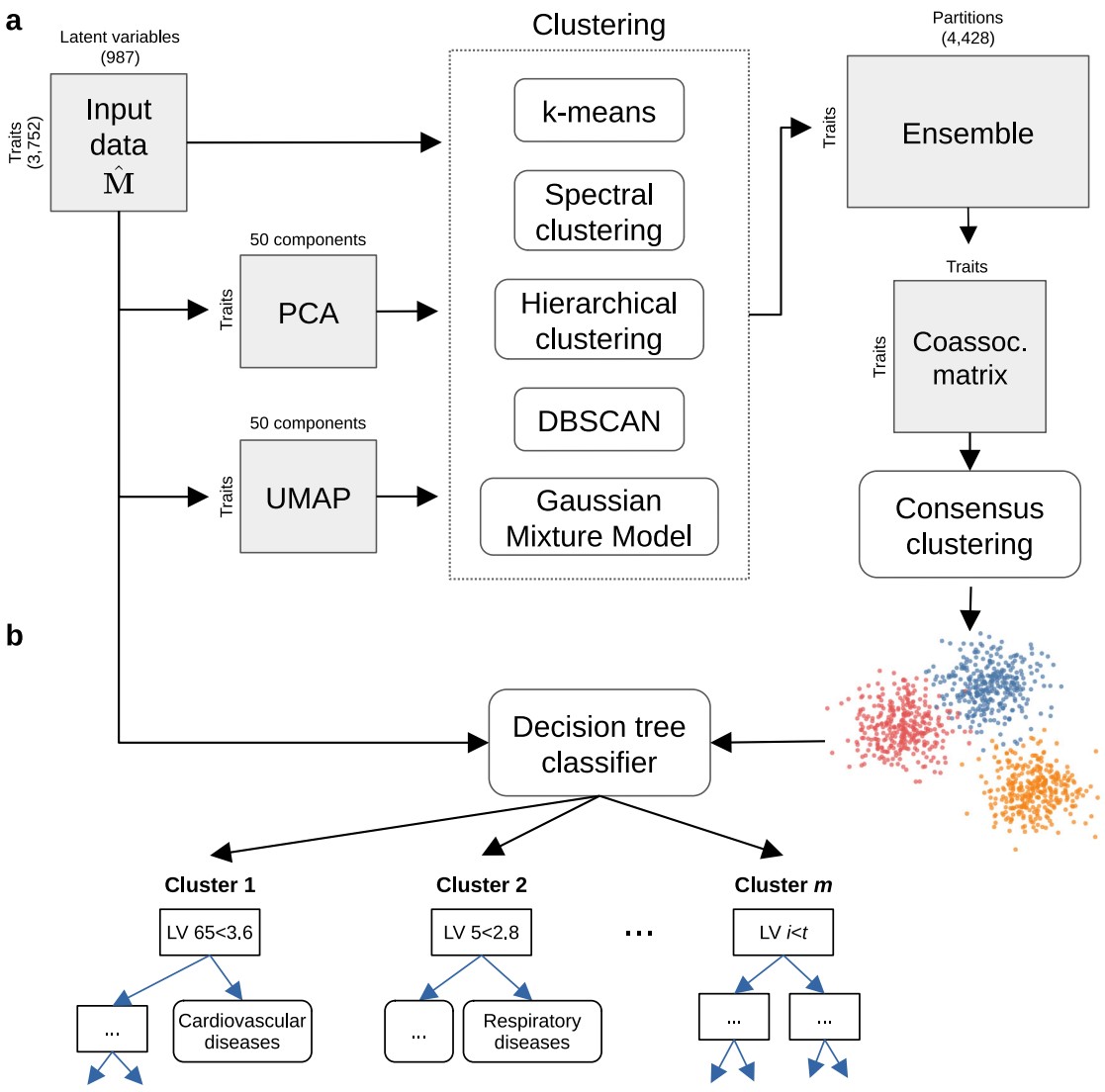

**Fig. 5 | Cluster analysis on traits using the latent gene expression representation. a** The projection of TWAS results on 3752 traits into the latent gene expression representation is the input data to the clustering process. A linear (PCA) and nonlinear (UMAP) dimensionality reduction techniques were applied to the input data, and five different clustering algorithms processed all data versions. These algorithms derive partitions from the data using different parameters (such as the number of clusters), leading to an ensemble of 4428 partitions. Then, a distance matrix is derived by counting how many times a pair of traits was grouped in different clusters across the ensemble. Finally, a consensus function is applied to the distance matrix to generate consolidated partitions with different numbers of clusters (from 2 to $\sqrt{n} \approx 60$). These final solutions were represented in the clustering tree (Fig. 6). **b** The clusters found by the consensus function were used as labels to train a decision tree classifier on the original input data, which detects the LVs that better differentiate groups of traits.

potentially involved in more general biological functions. We used our regression framework to determine whether these LVs were significantly associated with different traits. For example, LVs such as LV928 and LV30, which were well-aligned to early progenitors of the erythrocytes lineage[82] (Supplementary Tables 12 and 15), were predominantly expressed in early differentiation stages of erythropoiesis (Supplementary Figures 17 and 18) and strongly associated with different assays on red blood cells (FDR < 0.05; Supplementary Tables 13, 14, and 17). In contrast, other LVs were highly specific, such as LV730, which is expressed in thrombocytes from different cancer samples (Supplementary Figures 19 and Supplementary Table 18), and strongly associated with hematological assays on platelets (FDR < 0.05, Supplementary Table 19); or LV598, whose genes were expressed in corneal endothelial cells (Supplementary Figures 20 and Supplementary Table 21) and associated with keratometry measurements (Supplementary Table 22).

The sub-branches of autoimmune and cardiovascular diseases merged together at $k = 10$ (middle of Fig. 6), so we expected to find LVs

that specifically affect one or both of these types of diseases. For example, LV57, expressed in T cells (Supplementary Figure 21 and Supplementary Table 24), was the most strongly associated gene module with autoimmune disorders in PhenomeXcan (Supplementary Table 25), with significant associations with hypothyroidism that were replicated in eMERGE (Supplementary Table 26). However, this LV was also strongly associated with deep venous thrombosis in both PhenomeXcan and eMERGE. On the other hand, LV844 was more autoimmune-specific, with associations to polymyalgia rheumatica, type 1 diabetes, rheumatoid arthritis, and celiac disease in PhenomeXcan (Supplementary Table 28). However, these did not replicate in eMERGE. This LV was expressed in a wide range of cell types, including blood, breast organoids, myeloma cells, lung fibroblasts, and different cell types from the brain (Supplementary Figure 22 and Supplementary Table 27).

The cardiovascular sub-branch had 129 significant LV-trait associations in PhenomeXcan and 23 in eMERGE. LV136, aligned with known collagen formation and muscle contraction pathways

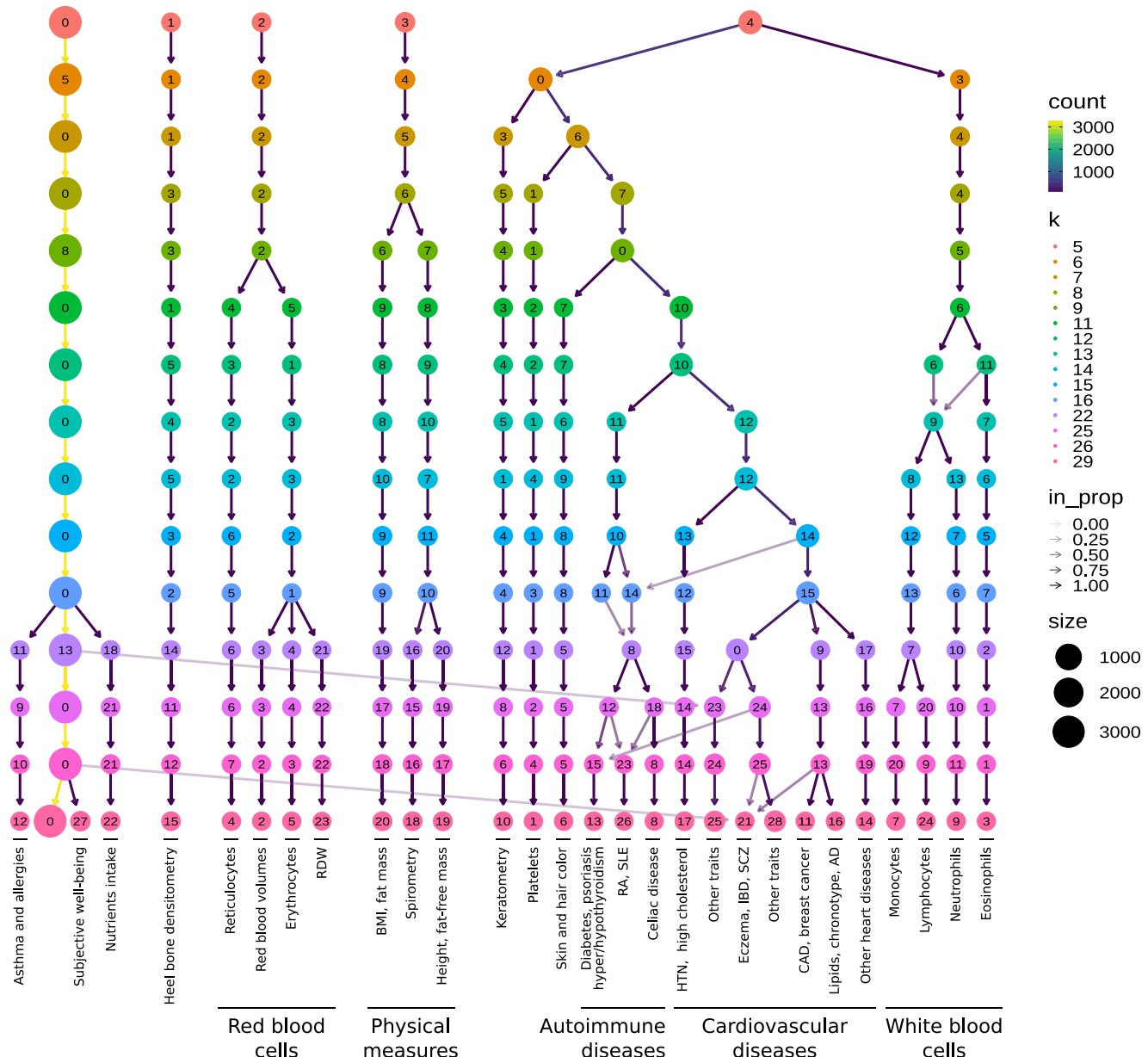

**Fig. 6 | Clustering tree using multiple resolutions for clusters of traits.** Each row represents a partition/grouping of the traits, and each circle is a cluster from that partition. The number of clusters goes from 5 to 29. Arrows indicate how traits in one cluster move across clusters from different partitions. Most of the clusters are preserved across different resolutions, showing highly stable solutions even with independent runs of the clustering algorithm. RDW red cell (erythrocyte) distribution width; BMI body mass index; WC waist circumference; HC hip circumference; RA rheumatoid arthritis; SLE systemic lupus erythematosus; HTN Hypertension; IBD inflammatory bowel disease; SCZ Schizophrenia; CAD Coronary artery disease; AD Alzheimer's disease; The full lists of traits in each cluster in the last five partitions of the tree (from $k = 16$ to $k = 29$) are in Supplementary Data 3-7.

(Supplementary Table 30), was associated with coronary artery disease and keratometry measurements in PhenomeXcan (Supplementary Table 31). In eMERGE, this LV was associated with coronary atherosclerosis (phecode: 411.4) (Supplementary Table 32). LV136 was expressed in a wide range of cell types, including fibroblasts, mesenchymal stem cells, osteoblasts, pancreatic stellate cells, cardiomyocytes, and adipocytes (Supplementary Figure 23). Within the cardiovascular sub-branch, we found neuropsychiatric and neurodevelopmental disorders such as Alzheimer's disease, schizophrenia, and attention deficit hyperactivity disorder (ADHD). These disorders were previously linked to the cardiovascular system[83–86] and share several risk factors, including hypertension, high cholesterol, obesity, smoking, among others[87,88]. However, our results grouped these diseases by potentially shared transcriptional processes expressed in specific tissues/cell types. Alzheimer's disease (not present in eMERGE), for instance, was significantly associated with LV21 in PhenomeXcan (Supplementary Table 34). LV21, a gene module not aligned to prior pathways, was strongly expressed in a variety of soft tissue sarcomas, monocytes/macrophages (including microglia from cortex samples), and aortic valves (Supplementary Figure 24 and Supplementary Table 33). This LV was also strongly associated with lipids and high cholesterol in PhenomeXcan and hyperlipidemia (phecode: 272.1) in eMERGE (Supplementary Table 35). As discussed previously, macrophages play a key role in the reverse cholesterol transport and thus atherogenesis[89], and lipid metabolism in microglia have been recently identified as an important factor in the development of neurodegenerative diseases[90].

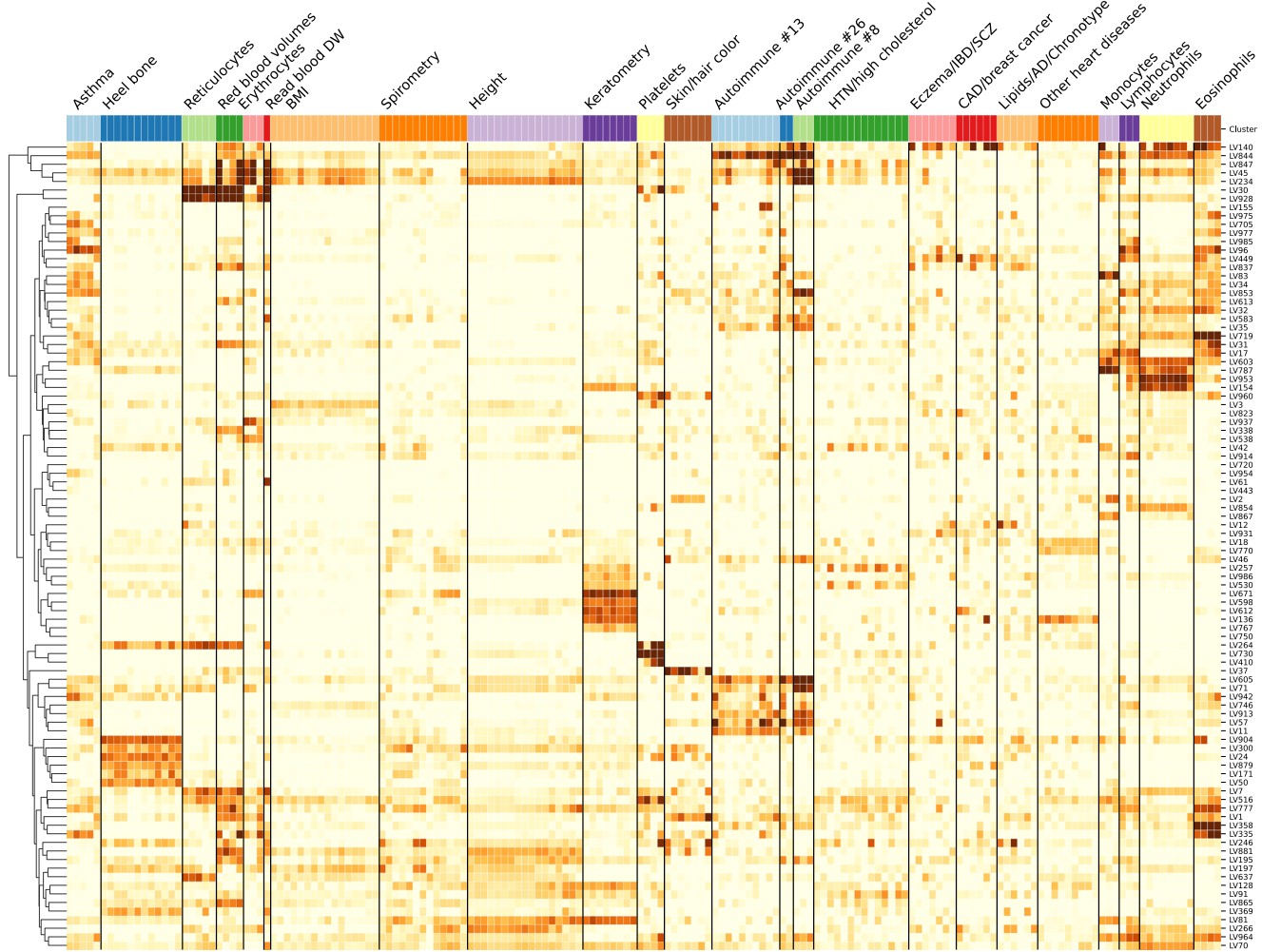

**Fig. 7 | Cluster-specific and general transcriptional processes associated with different diseases.** The plot shows a submatrix of $\hat{M}$ for the main trait clusters at $k = 29$, considering only LVs (rows) that are well-aligned with at least one pathway.

## Discussion

We have introduced a novel computational strategy that integrates statistical associations from TWAS with groups of genes (gene modules) that have similar expression patterns across the same cell types. Our key innovation is that we project gene-trait associations through a latent representation derived not strictly from measures of normal tissue but also from cell types under a variety of stimuli and at various developmental stages. This improves interpretation by going beyond statistical associations to infer cell type-specific features of complex phenotypes. Our approach can identify disease-relevant cell types from summary statistics, and several disease-associated gene modules were replicated in eMERGE. Using a CRISPR screen to analyze lipid regulation, we found that our gene module-based approach can prioritize causal genes even when single-gene associations are not detected. We interpret these findings with an omnigenic perspective of "core" and "peripheral" genes, suggesting that the approach can identify genes that directly affect the trait with no mediated regulation of other genes and thus prioritize alternative and potentially more attractive therapeutic targets.

Using our gene module perspective, we also integrated drug-induced transcriptional profiles, which allowed us to connect diseases, drugs, and cell types. We showed that the LV-based drug-repurposing approach outperformed the gene-based one when predicting drug-disease links for 322 drugs across 53 diseases. Furthermore, and beyond statistical prediction, we focused on cardiovascular traits and a particular drug, niacin, to show that the approach connects

pathophysiological processes with known mechanisms of action, including those in adipose tissue, immune cells, and ovarian granulosa cells. Our LV-based approach could be helpful in generating novel hypotheses to evaluate potential mechanisms of action, or even adverse effects, of known or experimental drugs.

We found that the analysis of associations through latent representations provided reasonable groupings of diseases and traits affected by shared and distinct transcriptional mechanisms expressed in highly relevant tissues. Our cluster analysis approach also detected the LVs that were most discriminative for each cluster. Several of these LVs were also significantly associated with different traits. Some LVs were strongly aligned with known pathways, but others (like LV57) were not, which might represent novel disease-relevant mechanisms. In some cases, the features/LVs linked to phenotypes appear to be associated with specific cell types. Associations with such cell type marker genes may reveal potentially causal cell types for a phenotype with more precision. We observed modules expressed primarily in one tissue (such as adipose in LV246 or ovary in LV66). Others appeared to be expressed in many contexts, which may capture pathways associated with related complex diseases. For example, LV136 is associated with cardiovascular disease and measures of corneal biomechanics and is expressed in fibroblasts, osteoblasts, pancreas, liver, and cardiomyocytes, among others. Other examples include LV844, expressed in whole blood samples and associated with a range of autoimmune diseases; or LV57, which is clearly expressed in T cells and strongly associated with autoimmune and venous thromboembolism.

From an omnigenic point of view, these patterns might represent cases of "network pleiotropy," where the same cell types mediate molecularly related traits. To our knowledge, projection through a representation learned on complementary but distinct datasets is a novel approach to identifying cell type and pathway effects on complex phenotypes that is computationally simple to implement.

We also demonstrated that clustering trees, introduced initially as a means to examine developmental processes in single-cell data, provide a multi-resolution grouping of phenotypes based on latent variable associations. We employed hard-partitioning algorithms (one trait belongs exclusively to one cluster) where the distance between two traits takes into account all gene modules. However, it is also plausible for two complex diseases to share only a few biological processes instead of being similar across most of them. Another important consideration is that our TWAS results were derived from a large set of GWAS of different sample sizes and qualities. Although the potential issues derived from this data heterogeneity were addressed before performing our cluster analyses on traits, data preprocessing steps are always challenging and might not avoid bias altogether. Considering groups of related diseases was previously shown to be more powerful in detecting shared genetic etiology[91,92], and clustering trees provide a way to explore such relationships in the context of latent variables.

Finally, we developed an LV-based regression framework to detect whether gene modules are associated with a trait using TWAS $p$-values. We used PhenomeXcan as a discovery cohort across four thousand traits, and many LV-trait associations replicated in eMERGE. In PhenomeXcan, we found 3450 significant LV-trait associations (FDR < 0.05) with 686 LVs (out of 987) associated with at least one trait and 1176 traits associated with at least one LV. In eMERGE, we found 196 significant LV-trait associations, with 116 LVs associated with at least one trait/phecode and 81 traits with at least one LV. We only focused on a few disease types from our trait clusters, but the complete set of associations on other disease domains is available in our Github repository for future research. As noted in Methods, one limitation of the regression approach is that the gene-gene correlations are only approximately accurate, which could lead to false positives if the correlation among the top genes in a module is not precisely captured. The regression model, however, is approximately well-calibrated, and we did not observe inflation when running the method in real data.

Our approach rests on the assumption that gene modules with coordinated expression patterns will also manifest coordinated pathological effects. Our implementation in this work integrates two complementary approaches. The first is MultiPLIER, which extracts latent variables from large expression datasets, and these LVs could represent either real transcriptional processes or technical factors ("batch effects"). We used a previously published model derived from recount2, which was designed to analyze rare disorders but might not be the optimal latent representation for the wide range of complex diseases considered here. Also, the underlying factorization method rests on linear combinations of variables, which could miss important and more complex co-expression patterns. In addition, recount2, the training dataset used, has since been surpassed in size and scale by other resources[20,93]. However, it is important to note that our models impose very few assumptions on the latent expression representation. Therefore, we should be able to easily replace MultiPLIER with other similar approaches like GenomicSuperSignature[94]. The second approach we used in this study is TWAS, where we are only considering the hypothesis that GWAS loci affect traits via changes in gene expression. Other effects, such as coding variants disrupting protein-protein interactions, are not captured. Additionally, TWAS has several limitations that can lead to false positives[95,96]. Like GWAS, which generally detects groups of associated variants in linkage disequilibrium (LD), TWAS usually identifies several genes within the same locus[25,97].

This is due to sharing of GWAS variants in gene expression models, correlated expression of nearby genes, or even correlation of their predicted expression due to eQTLs in LD, among others[95]. Our LV-based regression framework, however, accounts for these gene-gene correlations in TWAS reasonably well.

Our findings are concordant with previous studies showing that drugs with genetic support are more likely to succeed through the drug development pipeline[7,30]. In this case, projecting association results through latent variables better-prioritized disease-treatment pairs than considering single-gene effects alone. An additional benefit is that the latent variables driving predictions represent interpretable genetic features that can be examined to infer potential mechanisms of action. Here we prioritized drugs for diseases with very different tissue etiologies, and a challenge of the approach is to select the most appropriate tissue model from TWAS to find reversed transcriptome patterns between genes and drug-induced perturbations.

Ultimately, the quality of the representations is essential to performance. Here we used a representation derived from a factorization of bulk RNA-seq data. Detailed perturbation datasets and single-cell profiling of tissues, with and without perturbagens, and at various stages of development provide an avenue to generate higher quality and more interpretable representations. On the other hand, the key to interpretability is driven by the annotation of sample metadata. New approaches to infer and annotate with structured metadata are promising and can be directly applied to existing data[98]. Rapid improvements in both areas set the stage for latent variable projections to be widely applied to disentangle the genetic basis of complex human phenotypes. By providing a new perspective for a mechanistic understanding of statistical associations from TWAS, our method can generate testable hypotheses for the post-GWAS functional characterization of complex diseases, which will likely be an area of great importance in the coming years.

## Methods

PhenoPLIER is a framework that combines different computational approaches to integrate gene-trait associations and drug-induced transcriptional responses with groups of functionally related genes (referred to as gene modules or latent variables/LVs). Gene-trait associations are computed using the PrediXcan family of methods, whereas latent variables are inferred by the MultiPLIER models applied on large gene expression compendia. PhenoPLIER provides (1) a regression model to compute an LV-trait association, (2) a consensus clustering approach applied to the latent space to learn shared and distinct transcriptomic properties between traits, and (3) an interpretable, LV-based drug-repurposing framework. We provide the details of these methods below.

### The PrediXcan family of methods for gene-based associations

We used Summary-PrediXcan (S-PrediXcan)[99] and Summary-MultiXcan (S-MultiXcan)[24] as the gene-based statistical approaches, which belong to the PrediXcan family of methods[25]. We broadly refer to these approaches as TWAS (transcription-wide association studies). S-PrediXcan, the summary-based version of PrediXcan, computes the univariate association between a trait and a gene's predicted expression in a single tissue. In contrast, S-MultiXcan, the summary-based version of MultiXcan, computes the joint association between a gene's predicted expression in all tissues and a trait. S-PrediXcan and S-MultiXcan only need GWAS summary statistics instead of individual-level genotype and phenotype data.

Here we briefly provide the details about these TWAS methods that are necessary to explain our regression framework later (see the referenced articles for more information). In the following, we refer to $\mathbf{y}$ as a vector of traits for $n$ individuals that is centered for convenience (so that no intercept is necessary); $\tilde{\mathbf{t}}_l = \sum_{a \in \text{model}_l} w_a^l X_a$ is the gene's predicted expression for all individuals in tissue $l$, $X_a$ is the genotype of

SNP $a$ and $w_a$ its weight in the tissue prediction model $l$; and $\mathbf{t}_l$ is the standardized version of $\tilde{\mathbf{t}}_l$ with mean equal to zero and standard deviation equal to one.

S-PrediXcan[99] is the summary version of PrediXcan[25]. PrediXcan models the trait as a linear function of the gene's expression on a single tissue using the univariate model

$$\mathbf{y} = \mathbf{t}_l \gamma_l + \boldsymbol{\epsilon}_l, \tag{1}$$

where $\hat{\gamma}_l$ is the estimated effect size or regression coefficient, and $\boldsymbol{\epsilon}_l$ are the error terms with variance $\sigma_e^2$. The significance of the association is assessed by computing the $z$-score $\hat{z}_l = \hat{\gamma}_l/\text{se}(\hat{\gamma}_l)$ for a gene's tissue model $l$. PrediXcan needs individual-level data to fit this model, whereas S-PrediXcan approximates PrediXcan $z$-scores using only GWAS summary statistics with the expression

$$\hat{z}_l \approx \sum_{a \in model_l} w_a^l \frac{\hat{\sigma}_a}{\hat{\sigma}_l} \frac{\hat{\beta}_a}{\text{se}(\hat{\beta}_a)}, \tag{2}$$

where $\hat{\sigma}_a$ is the variance of SNP $a$, $\hat{\sigma}_l$ is the variance of the predicted expression of a gene in tissue $l$, and $\hat{\beta}_a$ is the estimated effect size of SNP $a$ from the GWAS. In these TWAS methods, the genotype variances and covariances are always estimated using the Genotype-Tissue Expression project (GTEx v8)[4] as the reference panel. Since S-PrediXcan provides tissue-specific direction of effects (for instance, whether a higher or lower predicted expression of a gene confers more or less disease risk), we used the $z$-scores in our drug-repurposing approach (described below).

S-MultiXcan[24], on the other hand, is the summary version of MultiXcan. MultiXcan is more powerful than PrediXcan in detecting gene-trait associations, although it does not provide the direction of effects. Its main output is the $p$-value (obtained with an $F$ test) of the multiple tissue model

$$\begin{aligned} \mathbf{y} &= \sum_{l=1}^{p} \mathbf{t}_l g_l + \mathbf{e} \\ &= \mathbf{T}\mathbf{g} + \mathbf{e}, \end{aligned} \tag{3}$$

where $\mathbf{T}$ is a matrix with $p$ columns $\mathbf{t}_l$, $\hat{g}_l$ is the estimated effect size for the predicted gene expression in tissue $l$ (and thus $\hat{\mathbf{g}}$ is a vector with $p$ estimated effect sizes $\hat{g}_l$), and $\mathbf{e}$ are the error terms with variance $\sigma_e^2$. Given the high correlation between predicted expression values for a gene across different tissues, MultiXcan uses the principal components (PCs) of $\mathbf{T}$ to avoid collinearity issues. S-MultiXcan derives the joint regression estimates (effect sizes and their variances) in Equation (3) using the marginal estimates from S-PrediXcan in Equation (2). Under the null hypothesis of no association, $\hat{\mathbf{g}}^\top \frac{\mathbf{T}^\top \mathbf{T}}{\sigma_e^2} \hat{\mathbf{g}} \sim \chi_p^2$, and therefore the significance of the association in S-MultiXcan is estimated with

$$\begin{aligned} \frac{\hat{\mathbf{g}}^\top (\mathbf{T}^\top \mathbf{T}) \hat{\mathbf{g}}}{\sigma_e^2} &\approx \hat{\boldsymbol{\gamma}}^\top \frac{\sqrt{n-1}}{\sigma_\epsilon} \left( \frac{\mathbf{T}^\top \mathbf{T}}{n-1} \right)^{-1} \frac{\sqrt{n-1}}{\sigma_\epsilon} \hat{\boldsymbol{\gamma}} \\ &= \hat{\mathbf{z}}^\top Cor(\mathbf{T})^{-1} \hat{\mathbf{z}}, \end{aligned} \tag{4}$$

where $\hat{\mathbf{z}}$ is a vector with $p$ $z$-scores (Equation (2)) for each tissue available for the gene, and $Cor(\mathbf{T})$ is the autocorrelation matrix of $\mathbf{T}$. Since $\mathbf{T}^\top \mathbf{T}$ is singular for many genes, S-MultiXcan computes the pseudo-inverse $Cor(\mathbf{T})^+$ using the $k$ top PCs, and thus $\hat{\mathbf{z}}^\top Cor(\mathbf{T})^+ \hat{\mathbf{z}} \sim \chi_k^2$. To arrive at this expression, S-MultiXcan uses the conservative approximation $\sigma_e^2 \approx \sigma_\epsilon^2$, that is, the variance of the error terms in the joint regression is approximately equal to the residual variance of the marginal regressions. Another important point is that $Cor(\mathbf{T})$ is estimated using a global genotype covariance matrix, whereas

marginal $\hat{z}_l$ in Equation (2) are approximated using tissue-specific genotype covariances. Although S-MultiXcan yields highly concordant estimates compared with MultiXcan, results are not perfectly correlated across genes[24]. As we explain later, these differences are important for our LV-based regression model when computing the gene-gene correlation matrix. We used S-MultiXcan results for our LV-based regression model and our cluster analyses of traits.

## TWAS resources

We used two large TWAS resources from different cohorts for discovery and replication, all obtained from European ancestries. PhenomeXcan[42], our discovery cohort, provides results on 4,091 traits across different categories. Supplementary Data 1 has all the details about the included GWAS, sample size, and disease/trait categories. In PhenomeXcan, these publicly available GWAS summary statistics were used to compute (1) gene-based associations with the PrediXcan family of methods (described before), and (2) a posterior probability of colocalization between GWAS loci and cis-eQTL with fastENLOC[42,96]. We refer to the matrix of $z$ scores from S-PrediXcan (Equation (2)) across $q$ traits and $m$ genes in tissue $t$ as $\mathbf{M}^t \in \mathbb{R}^{q \times m}$. As explained later, matrices $\mathbf{M}^t$ were used in our LV-based drug-repurposing framework since they provide direction of effects. The S-MultiXcan results (22,515 gene associations across 4,091 traits) were used in our LV-based regression framework and our cluster analyses of traits. For the cluster analyses, we used the $p$-values converted to $z$-scores: $\mathbf{M} = \Phi^{-1}(1-p/2)$, where $\Phi^{-1}$ is the probit function. Higher $z$-scores correspond to stronger associations.

Our discovery cohort was eMERGE[46], where the same TWAS methods were run on 309 phecodes[27] across different categories (more information about traits is available in ref. [27]). We used these results to replicate the associations found with our LV-based regression framework in PhenomeXcan.

## MultiPLIER and pathway-level information extractor (PLIER)

MultiPLIER[44] extracts patterns of co-expressed genes from recount2[19] (without including GTEx samples), a large gene expression dataset. The approach applies the PLIER[45], which performs unsupervised learning using prior knowledge (canonical pathways) to reduce technical noise. PLIER uses a matrix factorization approach that deconvolutes gene expression data into a set of latent variables (LV), where each LV represents a gene module. The MultiPLIER models reduced the dimensionality in recount2 to 987 LVs.

Given a gene expression dataset $\mathbf{Y}^{m \times c}$ with $m$ genes and $c$ experimental conditions and a prior knowledge matrix $\mathbf{C} \in \{0,1\}^{m \times p}$ for $p$ MSigDB pathways[100] (so that $\mathbf{C}_{ij} = 1$ if gene $i$ belongs to pathway $j$), PLIER finds $\mathbf{U}$, $\mathbf{Z}$, and $\mathbf{B}$ minimizing

$$||\mathbf{Y} - \mathbf{Z}\mathbf{B}||_F^2 + \lambda_1 ||\mathbf{Z} - \mathbf{C}\mathbf{U}||_F^2 + \lambda_2 ||\mathbf{B}||_F^2 + \lambda_3 ||\mathbf{U}||_{L^1} \tag{5}$$

subject to $\mathbf{U} > 0$, $\mathbf{Z} > 0$; $\mathbf{Z}^{m \times l}$ are the gene loadings with $l$ latent variables, $\mathbf{B}^{l \times c}$ is the latent space for $c$ conditions, $\mathbf{U}^{p \times l}$ specifies which of the $p$ prior-information pathways in $\mathbf{C}$ are represented for each LV, and $\lambda_i$ are different regularization parameters used in the training step. $\mathbf{Z}$ is a low-dimensional representation of the gene space where each LV aligns as much as possible with prior knowledge, and it might represent either a known or novel gene module (i.e., a meaningful biological pattern) or noise.

For our drug-repurposing and cluster analyses, we used this model to project gene-trait (from TWAS) and gene-drug associations (from LINCS L1000) into this low-dimensional gene module space. For instance, TWAS associations $\mathbf{M}$ (either from S-PrediXcan or S-MultiXcan) were projected using

$$\hat{\mathbf{M}} = (\mathbf{Z}^\top \mathbf{Z} + \lambda_2 \mathbf{I})^{-1} \mathbf{Z}^\top \mathbf{M}, \tag{6}$$

where $\hat{\mathbf{M}}^{l \times q}$ is a matrix where traits are represented by gene modules instead of single genes. As explained later, we used the same approach to project drug-induced transcriptional profiles in LINCS L1000 to obtain a representation of drugs using gene modules.

### Regression model for LV-trait associations

We adapted the gene set analysis framework from MAGMA[101] to TWAS. We used a competitive test to predict gene-trait associations from TWAS using gene weights from an LV, testing whether top-weighted genes for an LV are more strongly associated with the phenotype than other genes with relatively small or zero weights. Thus, we fit the model

$$\mathbf{m} = \beta_0 + \mathbf{s}\beta_s + \sum_i \mathbf{x}_i\beta_i + \boldsymbol{\epsilon}, \qquad (7)$$

where $\mathbf{m}$ is a vector of S-MultiXcan gene $p$-values for a trait (with a $-log_{10}$ transformation); $\mathbf{s}$ is a binary indicator vector with $s_\ell = 1$ for the top 1% of genes with the largest loadings for LV $\ell$ (from $\mathbf{Z}_\ell$) and zero otherwise; $\mathbf{x}_i$ is a gene property used as a covariate; $\beta$ are effect sizes (with $\beta_0$ as the intercept); and $\boldsymbol{\epsilon} \sim MVN(0, \sigma^2\mathbf{R})$ is a vector of error terms with a multivariate normal distribution (MVN) where $\mathbf{R}$ is the matrix of gene correlations.

The model tests the null hypothesis $\beta_s = 0$ against the one-sided hypothesis $\beta_s > 0$. Therefore, $\beta_s$ reflects the difference in trait associations between genes that are part of LV $\ell$ and genes outside of it. Following the MAGMA framework, we used two gene properties as covariates: (1) *gene size*, defined as the number of PCs retained in S-MultiXcan, and (2) *gene density*, defined as the ratio of the number of PCs to the number of tissues available.

Since the error terms $\boldsymbol{\epsilon}$ could be correlated, we cannot assume they have independent normal distributions as in a standard linear regression model. In the PrediXcan family of methods, the predicted expression of a pair of genes could be correlated if they share eQTLs or if these are in LD[95]. Therefore, we used a generalized least squares approach to account for these correlations. The gene-gene correlation matrix $\mathbf{R}$ was approximated by computing the correlations between the model sum of squares (SSM) for each pair of genes under the null hypothesis of no association. These correlations are derived from the individual-level MultiXcan model (Equation (3)), where the predicted expression matrix $\mathbf{T}_i \in \mathbb{R}^{n \times p_i}$ of a gene $i$ across $p_i$ tissues is projected into its top $k_i$ PCs, resulting in matrix $\mathbf{P}_i \in \mathbb{R}^{n \times k_i}$. From the MAGMA framework, we know that the SSM for each gene is proportial to $\mathbf{y}^\top \mathbf{P}_i \mathbf{P}_i^\top \mathbf{y}$. Under the null hypothesis of no association, the covariances between the SSM of genes $i$ and $j$ is therefore given by $2 \times \text{Trace}(\mathbf{P}_i^\top \mathbf{P}_j \mathbf{P}_j^\top \mathbf{P}_i)$. The standard deviations of each SSM are given by $\sqrt{2 \times k_i} \times (n-1)$. Therefore, the correlation between the SSMs for genes $i$ and $j$ can be written as follows:

$$\begin{aligned}\mathbf{R}_{ij} &= \frac{2 \times \text{Tr}(\mathbf{P}_i^\top \mathbf{P}_j \mathbf{P}_j^\top \mathbf{P}_i)}{\sqrt{2 \times k_i} \times \sqrt{2 \times k_j} \times (n-1)^2} \\ &= \frac{2 \times \text{Tr}(Cor(\mathbf{P}_i, \mathbf{P}_j) \times Cor(\mathbf{P}_j, \mathbf{P}_i))}{\sqrt{2 \times k_i} \times \sqrt{2 \times k_j}},\end{aligned} \qquad (8)$$

where columns $\mathbf{P}$ are standardized, Tr is the trace of a matrix, and the cross-correlation matrix between PCs $Cor(\mathbf{P}_i, \mathbf{P}_j) \in \mathbb{R}^{k_i \times k_j}$ is given by

$$\begin{aligned}Cor(\mathbf{P}_i, \mathbf{P}_j) &= Cor(\mathbf{T}_i \mathbf{V}_i^\top \text{diag}(\lambda_i)^{-1/2}, \mathbf{T}_j \mathbf{V}_j^\top \text{diag}(\lambda_j)^{-1/2}) \\ &= \text{diag}(\lambda_i)^{-1/2} \mathbf{V}_i \left(\frac{\mathbf{T}_i^\top \mathbf{T}_j}{n-1}\right) \mathbf{V}_j^\top \text{diag}(\lambda_j)^{-1/2},\end{aligned} \qquad (9)$$

where $\frac{\mathbf{T}_i^\top \mathbf{T}_j}{n-1} \in \mathbb{R}^{p_i \times p_j}$ is the cross-correlation matrix between the predicted expression levels of genes $i$ and $j$, and columns of $\mathbf{V}_i$ and scalars $\lambda_i$ are the eigenvectors and eigenvalues of $\mathbf{T}_i$, respectively. S-MultiXcan keeps only the top eigenvectors using a condition number threshold of $\frac{\max(\lambda_i)}{\lambda_i} < 30$. To estimate the correlation of predicted expression levels for genes $i$ in tissue $k$ and gene $j$ in tissue $l$, $(\mathbf{t}_k^i, \mathbf{t}_l^j)$ ($\mathbf{t}_k^i$ is the $k$th column of $\mathbf{T}_i$), we used[24]

$$\begin{aligned}\frac{(\mathbf{T}_i^\top \mathbf{T}_j)_{kl}}{n-1} &= Cor(\mathbf{t}_k^i, \mathbf{t}_l^j) \\ &= \frac{Cov(\mathbf{t}_k, \mathbf{t}_l)}{\sqrt{\widehat{\text{var}}(\mathbf{t}_k)\widehat{\text{var}}(\mathbf{t}_l)}} \\ &= \frac{Cov(\sum_{a \in \text{model}_k} w_a^k X_a, \sum_{b \in \text{model}_l} w_b^l X_b)}{\sqrt{\widehat{\text{var}}(\mathbf{t}_k)\widehat{\text{var}}(\mathbf{t}_l)}} \\ &= \frac{\sum_{\substack{a \in \text{model}_k \\ b \in \text{model}_l}} w_a^k w_b^l Cov(X_a, X_b)}{\sqrt{\widehat{\text{var}}(\mathbf{t}_k)\widehat{\text{var}}(\mathbf{t}_l)}} \\ &= \frac{\sum_{\substack{a \in \text{model}_k \\ b \in \text{model}_l}} w_a^k w_b^l \Gamma_{ab}}{\sqrt{\widehat{\text{var}}(\mathbf{t}_k)\widehat{\text{var}}(\mathbf{t}_l)}},\end{aligned} \qquad (10)$$

where $X_a$ is the genotype of SNP $a$, $w_a^k$ is the weight of SNP $a$ for gene expression prediction in the tissue model $k$, and $\Gamma = \widehat{\text{var}}(\mathbf{X}) = (\mathbf{X} - \bar{\mathbf{X}})^\top (\mathbf{X} - \bar{\mathbf{X}})/(n-1)$ is the genotype covariance matrix using GTEx v8 as the reference panel, which is the same used in all TWAS methods described here. The variance of the predicted expression values of gene $i$ in tissue $k$ is estimated as[99]:

$$\begin{aligned}\widehat{\text{var}}(\mathbf{t}_k^i) &= (\mathbf{W}^k)^\top \Gamma^k \mathbf{W}^k \\ &= \sum_{\substack{a \in \text{model}_k \\ b \in \text{model}_k}} w_a^k w_b^k \Gamma_{ab}^k.\end{aligned} \qquad (11)$$

Note that, since we used the MultiXcan regression model (Equation (3)), $\mathbf{R}$ is only an approximation of gene correlations in S-MultiXcan. As explained before, S-MultiXcan approximates the joint regression parameters in MultiXcan using the marginal regression estimates from S-PrediXcan in (2) with some simplifying assumptions and different genotype covariance matrices. This complicates the derivation of an S-MultiXcan-specific solution to compute $\mathbf{R}$. To account for this, we used a submatrix $\mathbf{R}_\ell$ corresponding to genes that are part of LV $\ell$ only (top 1% of genes) instead of the entire matrix $\mathbf{R}$. This simplification is conservative since correlations are accounted for top genes only. Our simulations (Supplementary Note 1) show that the model is approximately well-calibrated and can correct for LVs with adjacent and highly correlated genes at the top (e.g., Supplementary Figure 2). The simulation also identified 127 LVs in which the model was not well-calibrated (e.g., Supplementary Figure 6). As this can be attributed to limitations in accurately computing a gene correlation matrix, we excluded these LVs from our main analyses.

In Equation (10), for each gene, we only considered tissue models present in S-PrediXcan results, as well as SNPs present in GWAS used as input for the TWAS approaches. This is necessary to obtain more accurate correlation estimates[24]. Therefore, we computed different correlation matrices for PhenomeXcan and eMERGE. In PhenomeXcan, most of the GWAS (4049) were obtained from the UK Biobank using the same pipeline and including the same set of SNPs, so a single correlation matrix was used for this set. For the rest, we used a single correlation matrix for each group of traits that shared the same or most of the SNPs.

We ran our regression model for all 987 LVs across the 4091 traits in PhenomeXcan. For replication, we ran the model in the 309 phecodes in eMERGE. We adjusted the $p$-values using the Benjamini-Hochberg procedure.

## LV-based drug-repurposing approach

For the drug-disease prediction, we derived an LV-based method based on a drug repositioning framework previously used for psychiatry traits[30], where individual/single genes associated with a trait are anticorrelated with expression profiles for drugs. We compared our LV-based method with this previously published, single-gene approach. For the single-gene method, we computed a drug-disease score by multiplying each S-PrediXcan set of signed $z$-scores in tissue $t$, $\mathbf{M}^t$, with another set of signed $z$-scores from transcriptional responses profiled in LINCS L1000[43], $\mathbf{L}^{c \times m}$ (for $c$ compounds). Here $\mathbf{M}^t$ contains information about whether a higher or lower predicted expression of a gene is associated with disease risk, whereas $\mathbf{L}$ indicates whether a drug increases or decreases the expression of a gene. Therefore, these two matrices can be multiplied to compute a score for a drug-disease pair. The result of this product is $\mathbf{D}^{t,k} = -1 \cdot \mathbf{M}^{t,k} \mathbf{L}^\top$, where $k$ refers to the number of most significant gene associations in $\mathbf{M}^t$ for each trait. As suggested in[30], $k$ could be either all genes or the top 50, 100, 250, and 500; then, we averaged score ranks across all $k$ and obtained $\mathbf{D}^t$. Finally, for each drug-disease pair, we took the maximum prediction score across all tissues: $\mathbf{D}_{ij} = \max\{ijt\mathbf{D}|\forall t\}$.

The same procedure was used for the LV-based approach, where we projected $\mathbf{M}^t$ and $\mathbf{L}$ into the gene module latent space using Equation (6), leading to $\hat{\mathbf{M}}^t$ and $\hat{\mathbf{L}}^{l \times c}$, respectively. Finally, $\mathbf{D}^{t,k} = -1 \cdot \hat{\mathbf{L}}^\top \hat{\mathbf{M}}^{t,k}$, where in this case $k$ could be all LVs or the top 5, 10, 25, and 50 (since we have an order of magnitude less LVs than genes).

Since the gold standard of drug-disease medical indications is described with Disease Ontology IDs (DOID)[102], we mapped PhenomeXcan traits to the Experimental Factor Ontology[103] using[104], and then to DOID.

## Consensus clustering of traits

We performed two preprocessing steps on the S-MultiXcan results before the cluster analysis. First, we combined results in $\mathbf{M}$ (with $p$ values converted to $z$-scores, as described before) for traits that mapped to the same Experimental Factor Ontology (EFO)[103] term using the Stouffer's method: $\sum w_i M_{ij} / \sqrt{\sum w_i^2}$, where $w_i$ is a weight based on the GWAS sample size for trait $i$, and $M_{ij}$ is the $z$-score for gene $j$. Second, we divided all $z$-scores for each trait $i$ by their sum to reduce the effect of highly polygenic traits: $M_{ij}/\sum M_{ij}$. Finally, we projected this data matrix using Equation (6), obtaining $\hat{\mathbf{M}}$ with $n = 3752$ traits and $l = 987$ LVs as the input of our clustering pipeline.

A partitioning of $\hat{\mathbf{M}}$ with $n$ traits into $k$ clusters is represented as a label vector $\pi \in \mathbb{N}^n$. Consensus clustering approaches consist of two steps: (1) the generation of an ensemble $\Pi$ with $r$ partitions of the dataset: $\Pi = \{\pi_1, \pi_2, ..., \pi_r\}$, and (2) the combination of the ensemble into a consolidated solution defined as:

$$\pi^* = \arg\max_{\tilde{\pi}} Q(\{|\mathcal{L}^i|\phi(\hat{\pi}_{\mathcal{L}^i}, \pi_{i\mathcal{L}^i}) \mid i \in \{1, ..., r\}\}), \quad (12)$$

where $\mathcal{L}^i$ is a set of data indices with known cluster labels for partition $i$, $\phi : \mathbb{N}^n \times \mathbb{N}^n \to \mathbb{R}$ is a function that measures the similarity between two partitions, and $Q$ is a measure of central tendency, such as the mean or median. We used the adjusted Rand index (ARI)[105] for $\phi$ and the median for $Q$. To obtain $\pi^*$, we define a consensus function $\Gamma : \mathbb{N}^{n \times r} \to \mathbb{N}^n$ with $\Pi$ as the input. We used consensus functions based on the evidence accumulation clustering (EAC) paradigm[79], where $\Pi$ is first transformed into a distance matrix $\mathbf{D}_{ij} = d_{ij}/r$, where $d_{ij}$ is the number of times traits $i$ and $j$ were grouped in different clusters across all $r$ partitions in $\Pi$. Then, $\Gamma$ can be any similarity-based clustering algorithm, which is applied on $\mathbf{D}$ to derive the final partition $\pi^*$.

For the ensemble generation step, we used different algorithms to create a highly diverse set of partitions (see Fig. 5) since diversity is an important property for ensembles[106–108]. We used three data representations: the raw dataset, its projection into the top 50 principal components, and the embedding learned by UMAP[109] using 50 components. For each of these, we applied five clustering algorithms covering a wide range of different assumptions on the data structure: $k$-means[110], spectral clustering[111], a Gaussian mixture model (GMM), hierarchical clustering, and DBSCAN[112]. For $k$-means, spectral clustering, and GMM, we specified a range of $k$ between 2 and $\sqrt{n} \approx 60$, and for each $k$ we generated five partitions using random seeds. For hierarchical clustering, for each $k$, we generated four partitions using common linkage criteria: ward, complete, average, and single. For DBSCAN, we combined different ranges for parameters $\epsilon$ (the maximum distance between two data points to be considered part of the same neighborhood) and $minPts$ (the minimum number of data points in a neighborhood for a data point to be considered a core point), based on the procedure in[113]. Specifically, we used $minPts$ values from 2 to 125. For each data representation (raw, PCA, and UMAP), we determined a plausible range of $\epsilon$ values by observing the distribution of the mean distance of the $minPts$-nearest neighbors across all data points. Since some combinations of $minPts$ and $\epsilon$ might not produce a meaningful partition (for instance, when all points are detected as noisy or only one cluster is found), we resampled partitions generated by DBSCAN to ensure an equal representation of this algorithm in the ensemble. This procedure generated a final ensemble of 4428 partitions of 3752 traits.

Finally, we used spectral clustering on $\mathbf{D}$ to derive the final consensus partitions. $\mathbf{D}$ was first transformed into a similarity matrix by applying an RBF kernel $\exp(-\gamma \mathbf{D}^2)$ using four different values for $\gamma$ that we empirically determined to work best. Therefore, for each $k$ between 2 and 60, we derived four consensus partitions and selected the one that maximized Equation (12). We further filtered this set of 59 solutions to keep only those with an ensemble agreement larger than the 75th percentile (Supplementary Figure 13), leaving a total of 15 final consensus partitions shown in Fig. 6.

The input data in our clustering pipeline undergoes several linear and nonlinear transformations, including PCA, UMAP, and the ensemble transformation using the EAC paradigm (distance matrix $\mathbf{D}$). Although consensus clustering has clear advantages for biological data[77], this set of data transformations complicates the interpretation of results. To circumvent this, we used a supervised learning approach to detect which gene modules/LVs are the most important for each cluster of traits (Fig. 5b). Note that we did not use this supervised model for prediction but only to learn which features (LVs) were most discriminative for each cluster. For this, we used the highest resolution partition ($k = 29$, although any could be used) to train a decision tree model using each of the clusters as labels and the projected data $\hat{\mathbf{M}}$ as the training samples. For each $k$, we built a set of binary labels with the current cluster's traits as the positive class and the rest of the traits as the negative class. Then, we selected the LV in the root node of the trained model only if its threshold was positive and larger than one standard deviation. Next, we removed this LV from $\hat{\mathbf{M}}$ (regardless of being previously selected or not) and trained the model again. We repeated this procedure 20 times to extract the top 20 LVs that better discriminate traits in a cluster from the rest.

In Supplementary Note 2, we performed several analyses under a null hypothesis of no structure in the data to verify that the clustering results detected by this pipeline were real.

## CRISPR-Cas9 screening

**Cell culture.** HepG2 cells were obtained from ATCC (ATCC® HB-8065™), and maintained in Eagle's Minimum Essential Medium with L-Glutamine (EMEM, Cat. 112-018-101, Quality Biology) supplemented with 10% Fetal Bovine Serum (FBS, Gibco, Cat.16000-044), and 1% Pen/

Strep (Gibco, Cat.15140-122). Cells were kept at 37oC in a humidity-controlled incubator with 5% $CO_2$, and were maintained at a density not exceeding more than 80% confluency in Collagen-I coated flasks.

**Genome-wide lentiviral pooled CRISPR-Cas9 library.** 3rd lentiviral generation, Broad GPP genome-wide Human Brunello CRISPR knockout Pooled library was provided by David Root and John Doench from Addgene (Cat. 73179-LV), and was used for HepG2 cell transduction. It consists of 76,441 sgRNAs, and targets 19,114 genes in the human genome with an average of 4 sgRNAs per gene. Each 20nt sgRNA cassette was inserted into the lentiCRIS-PRv2 backbone between U6 promoter and gRNA scaffold. Through cell transduction, the lentiviral vectors that encode Cas9 were used to deliver the sgRNA cassette containing plasmids into cells during cell replication. Unsuccessful transduced cells were excluded through puromycin selection.

**Lentiviral titer determination.** No-spin lentiviral transduction was utilized for the screen. In a Collagen-I coated 6-wells plate, ~2.5 M cells were seeded each well in the presence of 8 μg/ml polybrene (Millipore Sigma, Cat. TR-1003 G), and a different titrated virus volume (e.g., 0, 50, 100, 200, 250, and 400 μl) was assigned to each well. EMEM complete media was added to make the final volume of 1.24ml. 16-18hrs post-transduction, virus/polybrene-containing media was removed from each well. Cells were washed twice with 1x DPBS and replaced with fresh EMEM. At 24h, cells in each well were trypsinized, diluted (e.g.,1:10), and seeded in pairs of wells of 6-well plates. At 60 hr post-transduction, cell media in each well was replaced with fresh EMEM. 2ug/ml of puromycin (Gibco, Cat. A1113803) was added to one well out of the pair. 2–5 days after puromycin selection, or the 0 virus well treated with puromycin had no survival of cells, cells in both wells with/without puromycin were collected and counted for viability. Percentage of Infection (PI%) was obtained by comparing the cell numbers with/without puromycin selection within each pair. By means of Poisson's distribution theory, when transduction efficiency (PI%) is between 30 and 50%, which corresponds to an MOI (Multiplicity of Infection) of ~0.35–0.70. At MOI close to 0.3, around 25% of cells are infected, and the majority of those infected cells are predicted to have only one copy of the virus. Therefore, a volume of virus (120ul) yielding 30–40% of transduction efficiency was chosen for further large-scale viral transduction.

**Lentiviral Transduction in HepG2 Using Brunello CRISPR Knockout Pooled Library.** In order to achieve a coverage (representation) of at least 500 cells per sgRNA, and at an MOI between 0.3–0.4 to ensure 95% of infected cells get only one viral particle per cell, ~200 M cells were initiated for the screen. Transduction was carried out in a similar fashion as described above. Briefly, 2.5 M cells were seeded in each well of 14 6-well plates, along with 8ug/ml of polybrene. A volume of 120 μl of the virus was added to each experimental well. 18hrs post-transduction, virus/PB mix medium was removed, and cells in each well were collected, counted, and pooled into T175 flasks. At 60hr post-transduction, 2ug/ml of puromycin was added to each flask. Mediums were changed every two days with fresh EMEM, topped with 2ug/ml puromycin. Seven days after puromycin selection, cells were collected, pooled, counted, and replated.

**Fluorescent dye staining.** 9 days after puromycin selection, cells were assigned to two groups. 20–30M cells were collected as Unsorted Control. The cell pellet was spun down at $500 \times g$ for 5 min at 4°C. The dry pellet was kept at −80 °C for further genomic DNA isolation. The rest of the cells (~200M) were kept in 100mm dishes and stained with a fluorescent dye (LipidSpotTM 488, Biotium, Cat. 70065-T). In Brief, LipidSpot 488 was diluted to 1:100 with DPBS. 4ml of staining solution was used for each dish and incubated at 37°C for 30 min. Cell images

were captured through fluorescent microscope EVOS for GFP signal detection (Supplementary Figure 8).

**Fluorescence-activated cell sorting (FACS).** Cells were immediately collected into 50ml tubes (From this point on, keep cells cold), and spun at $500 \times g$ for 5 min at 4°C. After DPBS wash, cell pellets were resuspended with FACS Sorting Buffer (1× DPBS without Ca2+/Mg2+, 2.5 mM EDTA, 25 mM HEPES, 1% BSA. The solution was filter sterilized, and kept at 4 °C), with gentle pipetting to make single cells. The cell solution was then filtered through a cell strainer (Falcon, Cat. 352235) and was kept on ice, protected from light. Collected cells were sorted on FACSJazz. 100 μm nozzle was used for sorting. ~20% of each GFP-High and GFP-Low (Supplementary Figure 9) were collected into 15ml tubes. After sorting, cells were immediately spun down. Pellets were kept at −80 °C for further genomic DNA isolation.

**Genomic DNA isolation and verification.** Three conditions of Genomic DNA (UnSorted Control, lentiV2 GFP-High, and lentiV2 GFP-Low) were extracted using QIAamp DNA Blood Mini Kit (Qiagen, Cat.51104), followed by UV Spectroscopy (Nanodrop) to access the quality and quantity of the gDNA. A total of 80–160 μg of gDNA was isolated for each condition. sgRNA cassette and lentiviral-specific transgene in isolated gDNA were verified through PCR (Supplementary Figure 10).

**Illumina libraries generation and sequencing.** The fragment containing sgRNA cassette was amplified using P5 /P7 primers, as indicated in[114], and primer sequences were adapted from Broad Institute protocol (Supplementary Figure 11). Stagger sequence (0–8nt) was included in P5 and 8bp uniquely barcoded sequence in P7. Primers were synthesized through Integrated DNA Technologies (IDT), and each primer was PAGE purified. 32 PCR reactions were set up for each condition. Each 100μ l PCR reaction consists of roughly 5 μg of gDNA, 5 μl of each 10 μM P5 and P7. ExTaq DNA Polymerase (TaKaRa, Cat. RR001A) was used to amplify the amplicon. PCR Thermal Cycler Parameters are set as Initial at 95°C for 1min; followed by 24 cycles of Denaturation at 94°C for 30 seconds, Annealing at 52.5 °C for 30 seconds, Extension at 72°C for 30 seconds. A final Elongation at 72°C for 10 minutes. 285bp–293bp PCR products were expected (Supplementary Figure 12A). PCR products within the same condition were pooled and purified using SPRIselect beads (Beckman Coulter, Cat. B23318). Purified Illumina libraries were quantitated on Qubit, and the quality of the library was analyzed on Bio-analyzer using a sensitivity DNA Chip. A single ~285bp peak was expected (Supplementary Figure 12B). Final Illumina library samples were sequenced on Nova-seq 6000. Samples were pooled and loaded on an SP flow cell, along with a 20% PhiX control v3 library spike-in.

### Reporting summary
Further information on research design is available in the Nature Portfolio Reporting Summary linked to this article.

## Data availability
All the main datasets generated in this study are available at https://doi.org/10.5281/zenodo.8071382[115] and the GitHub repository https://github.com/greenelab/phenoplier. The main input datasets used are TWAS from PhenomeXcan[42] for 4,091 traits and from the Electronic Medical Records and Genomics (eMERGE) network phase III[27] for 309 traits; transcriptional responses to small molecule perturbations from LINCS L1000[43] that were further preprocessed and mapped to Drug-Bank IDs from[54]; latent space/gene module models from MultiPLIER[44]. The data used from PhenomeXcan, LINCS L1000, and MultiPLIER are publicly available. All significant results reported for the eMERGE and Penn Medicine BioBank (PMBB) phenome-wide TWAS are contained in[27]. The individual-level PMBB raw datasets can not be made publicly available due to institutional privacy policy. Please contact Penn

Medicine Biobank (https://pmbb.med.upenn.edu/pmbb/) for requests for access to data. eMERGE network phase III data is available on dbGAP (Accession: phs001584.v2.p2).

## Code availability

The code necessary to reproduce all the analyses in this work is available at https://doi.org/10.5281/zenodo.8071382[115] and the GitHub repository https://github.com/greenelab/phenoplier. For the CRISPR screening, we used FlowJo v10.7 and FACSJazz Software v1.1. For data analysis, we used Python 3.8 and R 3.6 with several computational packages. The main Python packages used were: Jupyter Lab (2.2), pandas (1.1), matplotlib (3.3), seaborn (0.11), numpy (1.19), scipy (1.5), scikit-learn (0.23), and umap-learn (0.4). The main R packages were: Bioconductor (3.10), clusterProfiler (3.14), clustree (0.4), and fgsea (1.17). We also developed several scripts and notebooks which are published under an open-source license. We documented all the steps necessary to carry out all the analyses. We also provide a Docker image to use the same runtime environment we used, and a demo to quickly test the methods on real data.

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

## Acknowledgements

This study was funded by: the Gordon and Betty Moore Foundation (GBMF 4552 to C.S.G.; GBMF 4560 to B.D.S.), the National Human Genome Research Institute (R01 HG010067 to C.S.G., S.F.A. Grant, and B.D.S.; K99 HG011898 and R00 HG011898 to M. Pividori; U01 HG011181 to W.W.), the National Cancer Institute (R01 CA237170 to C.S.G.), the Eunice Kennedy Shriver National Institute of Child Health and Human Development (R01 HD109765 to C.S.G.), the National Institute of Aging (R01AG069900 to W.W.), the National Institute of General Medical Sciences (R01 GM139891 to W.W.); the National Heart, Lung, and Blood Institute (R01 HL163854 to Q.F.); the National Institute of Diabetes and Digestive and Kidney Diseases (DK126194 to B.F.V.); the Daniel B. Burke Endowed Chair for Diabetes Research to S.F.A. Grant; the Robert L. McNeil Jr. Endowed Fellowship in Translational Medicine and Therapeutics to C. Skarke. Phase III of the eMERGE Network was initiated and funded by the NHGRI through the following grants: U01 HG8657 (Group Health Cooperative/University of Washington); U01 HG8685 (Brigham and Womens Hospital); U01 HG8672 (Vanderbilt University Medical Center); U01 HG8666 (Cincinnati Childrens Hospital Medical Center); U01 HG6379 (Mayo Clinic); U01 HG8679 (Geisinger Clinic); U01 HG8680 (Columbia University Health Sciences); U01 HG8684 (Childrens Hospital of Philadelphia); U01 HG8673 (Northwestern University); U01 HG8701 (Vanderbilt University Medical Center serving as the Coordinating Center); U01 HG8676 (Partners Healthcare/Broad Institute); and U01 HG8664 (Baylor College of Medicine). The Penn Medicine BioBank (PMBB) is funded by the Perelman School of Medicine at the University of Pennsylvania, a gift from the Smilow family, and the National Center for Advancing Translational Sciences of the National Institutes of Health under CTSA Award Number UL1TR001878. We thank D. Birtwell, H. Williams, P. Baumann, and M. Risman for their informatics support regarding the PMBB. We thank the staff of the Regeneron Genetics

Center for whole-exome sequencing of DNA from PMBB participants. Figure 1a was created with BioRender.com.

## Author contributions

M.P. and C.S.G. conceived and designed the study. M.P. designed the computational methods, performed the experiments, analyzed the data, interpreted the results, and drafted the manuscript. C.S.G. supervised the entire project and provided critical guidance throughout the study. S.L., C.Su., and M.E.J. performed the CRISPR screen with the supervision of S.F.A.G. B.L. provided the TWAS results for eMERGE for replication, and this analysis was supervised by M.D.R. W.W., Q.F., B.N., K.K., I.K., Y.L., and M.D.R., as part of the eMERGE consortium, provided feedback regarding the analyses of this data. B.S., B.V., and C.Skarke. provided critical feedback.

## Competing interests

The authors declare no competing interests.

## Additional information

[1]Department of Genetics, Perelman School of Medicine, University of Pennsylvania, Philadelphia, PA 19104, USA. [2]Department of Biomedical Informatics, University of Colorado School of Medicine, Aurora, CO 80045, USA. [3]Center for Spatial and Functional Genomics, Children's Hospital of Philadelphia, Philadelphia, PA 19104, USA. [4]Department of Biomedical Data Science, Stanford University, Stanford, CA 94305, USA. [5]Vanderbilt University Medical Center, Nashville, TN 37232, USA. [6]Cincinnati Children's Hospital Medical Center, Cincinnati, OH 45229, USA. [7]Department of Medicine, Division of Nephrology, Vagelos College of Physicians & Surgeons, Columbia University, New York, NY 10032, USA. [8]Mayo Clinic, Rochester, MN 55905, USA. [9]Northwestern University, Chicago, IL 60611, USA. [10]Kahlert School of Computing, University of Utah, Salt Lake City, UT 84112, USA. [11]Department of Systems Pharmacology and Translational Therapeutics, Perelman School of Medicine, University of Pennsylvania, Philadelphia, PA 19104, USA. [12]Institute for Translational Medicine and Therapeutics, Perelman School of Medicine, University of Pennsylvania, Philadelphia, PA 19104, USA. [13]Division of Endocrinology and Diabetes, Children's Hospital of Philadelphia, Philadelphia, PA 19104, USA. [14]Division of Human Genetics, Children's Hospital of Philadelphia, Philadelphia, PA 19104, USA. [15]Department of Pediatrics, Perelman School of Medicine, University of Pennsylvania, Philadelphia, PA 19104, USA. [16]Center for Health AI, University of Colorado School of Medicine, Aurora, CO 80045, USA. ✉e-mail: casey.s.greene@cuanschutz.edu

## eMERGE Consortium

Wei-Qi Wei ⓘ [5], Qiping Feng ⓘ [5], Bahram Namjou ⓘ [6], Krzysztof Kiryluk[7], Iftikhar J. Kullo ⓘ [8], Yuan Luo ⓘ [9] & Marylyn D. Ritchie ⓘ [1]

A full list of members and their affiliations appears in the Supplementary Information.

