## [Peer Review File · Nature Communications]

Projecting genetic associations through gene expression patterns highlights disease etiology and drug mechanismsEditorial Note: Parts of this Peer Review File have been redacted as indicated to remove third-party material where no permission to publish could be obtained.

REVIEWER COMMENTS

Reviewer #1 (Remarks to the Author):

The manuscript by Pividori et al presents a set of integrative analyses that merge together gene-trait associations (from TWAS) with gene perturbation data to gain insights into which groups of genes are impacting which group of traits. The key insight of the approach is that gene modules (i.e. group of genes that share co-expression patterns) are more informative than modeling single gene entities. The manuscript is extremely densely written with many different analyses that are hard to follow, insufficient motivation for proposed analyses, and very few (if any) comparisons to alternate approaches to guide the parameter selection. Thus, I am moderately enthusiastic for the proposed manuscript.

I hope comments below are useful for the authors:

1. Core to the approach is the concept of gene modules (latent variables, LD) that are reconstructed within a matrix factorization approach (MultPLIER). How are the modules (LVs) robust to false network discoveries? E.g. recent works (Parsana et al Genome Biol 2019) note that regular technical/biological artifacts confound common gene expression modules with PC correction being able to reduce false discoveries. Is a PC correction performed here? Are the LV robust to different reconstruction methods?
2. An LV-trait association p-value is proposed to connect modules to traits. Is the p-value calibrated? I could not follow what is the null model rejected here; is that all genes in a module are not associated to the trait? At minimum the authors should present some null simulations to showcase that their proposed approach is not subjected to false positive rates.
3. The TWAS results integrated here are coming from a single TWAS method applied to UKBiobank data; first, the authors could investigate the robustness of their results to the PrediXscan method. But most importantly, given that the GWAS data utilize the same set of individuals will induce covariances in the TWAS statistics (due to shared environments) that will create spurious covariances in any linear combination of TWAS z-scores. The authors could investigate how these correlation may bias their approach.

4. What is the impact of using out-of-sample LD from 1000 Genomes instead of in-sample LD in equation 3? Recent works that prioritize TWAS z-scores to find causal genes note that such reference LD may bias inference (Mancuso et al Nat Genet 2019). The authors could easily use the UKBiobank LD to check robustness in their results.

5. In the clustering analysis, 5 clustering approaches are applied to the same data and a consensus is taken together. Same input data and similarities in the clustering algorithms will induce covariances in the clustering results; how are such expected covariances taken into account in the ensemble method? Second, the proposed ensemble approach may be overfitting the data; I suggest to do cross-validation outside(prior) to the application of 5 clustering approaches (in addition to the cross-validation within each clustering approach to select their best parameters).

6. I suggest the authors separate the discussion section to improve readability.

Reviewer #2 (Remarks to the Author):

In this work, the authors derived factorized co-expression gene modules weights from large-scale expression compendium by using previous MultiPLIER method. And they proposed a framework PhenoPLIER to test these co-expressed gene-trait associations and co-expressed gene-drug signature associations by utilizing PhenomeXcan and LINCS L1000 data, respectively. The authors identified a few significant associations and used CRISPR screen to validate one of their findings in lipid trait. Overall, it is a systematic design. However, this work heavily relies on their previous matrix factorization-based method MultiPLIER. The majority of the results are using the same analytical strategy to interpret the significant associations in latent variables (LVs) projections. The major innovation parts are the methodology of adding PhenomeXcan results and drug signature to the LVs and subsequent clustering analysis of the M^{\wedge} and decision tree on traits. So I have a few questions in terms of these two methods.

For the PhenoPLIER model:

1. Many regulatory SNPs and pleiotropic effect have population-specificity. TWAS and fastEnLOC results are also population sensitive. However, the gene expression compendium might not have the corresponding population information. Thus, how PhenoPLIER model could address the population background difference from gene expression and GWAS trait?

2. Why the model only consider the magnitude of S-MultiXcan without considering the effect direction when combining the TWAS effect to LV? Can the author give an explanation for choosing multiplicative model as the combining strategy of the weights instead of other models, such as additive model?

3. TWAS analysis either from S-PrediXcan or S-MultiXcan could only infer the association between GWAS traits and GTEx available tissues. What is the rationale by adding TWAS effect to the LVs of unrelated tissues or cell types factorized from gene expression compendium?

4. In the PhenoPLIER model, the S-PrediXcan results were projected to M^A is not well-described, whether they has direction or not. And how it will be different from the S-MultiXcan model.

For the Cluster analysis on traits method:

1. The method of using 5 clustering algorithms to measure the consensus clustering seem inspiring and could be expanded to other scenarios for the readers. Can the authors comments on the methods are the state-of-the-art method or not?

2. I wonder how robust these decision trees will be when the latent variables pool changes along with the accumulation of the gene expression compendium. On the other hand, I wonder what is the composition of the 3,752 traits? And how could they be clustered in only 29 groups with biological meaningful interpretation?

Reviewer #3 (Remarks to the Author):

In this manuscript, Pividoriet al. developed PhenoPLIER, a method that integrates TWAS datasets or drug profile data through latent variables (gene modules) and claims that using gene modules improves interpretability in the results.

Using MultiPLIER, latent variables are first learned from various bulk RNA-seq data (recount2) represented by gene modules and reduces dimensionality to 987 latent variables. Then, it uses TWAS data from PhenomeXcan and transcriptional profiles of drug data from LINCS L1000 and projects these into the same low-dimensional latent variables. Several examples were included: 1) identified diseases associated with gene modules such as hypothyroidism with T cells. 2) performed a CRISPR screen that showed more robust trait associations with gene modules than single genes. 3) drug disease prediction was better using gene modules than single genes. 4) clustering the projected TWAS data then training a decision tree classifier showed insights on transcriptional pathways.

However, using gene co-expression patterns to infer trait-relevant tissues is not novel in the field, such as CoCoNet (Leveraging gene co-expression patterns to infer trait-relevant tissues in genome-wide association studies, PLoS Genet. 2020). There are also many tools/papers (such as LDSC-SEG, deTS and RolyPoly, as below) using cell/tissue specific expressed genes to infer trait-relevant tissues. Comparing with these tools, I do not think PhenoPLIER has big improvement, maybe even worse. In the manuscript, the authors only compare PhenoPLIER results with TWAS results (it's a little strange because TWAS results are the input data of PhenoPLIER) but not with these similar tools. On the other aspect,

PhenoPLIER can expand TWAS genes with co-expression network, as example, the authors found that lipids-related genes DGAT2 and ACACA cannot be identified TWAS but PhenoPLIER. However, the authors did not analyze how many false positive genes (I guess a lot) can also be included by co-expression network. I am sure I can also find many functional trait-related genes by simply expanding TWAS risk genes with protein-protein interaction network or pathways. So, the DGAT2/ACACA example cannot convince me. BTW, the manuscript is not well-written. My detailed questions are listed as follows.

A list of tools/papers can infer trait-relevant tissues:

1. Shang L, Smith J A, Zhou X. Leveraging gene co-expression patterns to infer trait-relevant tissues in genome-wide association studies[J]. PLoS genetics, 2020, 16(4): e1008734.
2. Calderon D, Bhaskar A, Knowles DA, Golan D, Raj T, Fu AQ, et al. Inferring Relevant Cell Types for Complex Traits by Using Single-Cell Gene Expression. *Am J Hum Genet.* 2017;101(5):686–99. WOS:000414251600003. pmid:29106824
3. Finucane, H. K., Reshef, Y. A., Anttila, V., Slowikowski, K., Gusev, A., Byrnes, A., et al. (2018). Heritability enrichment of specifically expressed genes identifies disease-relevant tissues and cell types. *Nat. Genet.* 50, 621–629. doi: 10.1038/s41588-018-0081-4
4. Pei, G., Dai, Y., Zhao, Z., and Jia, P. (2019). deTS: tissue-specific enrichment analysis to decode tissue specificity. *Bioinformatics* 35, 3842–3845. doi: 10.1093/bioinformatics/btz138

Besides, this method is based on multiple existing methods. My concern about this multi-step approach is the error accumulation. Hence, a negative control is necessary.

Comments:

- Does the method and its results account for false positives? the results lack a "negative control." Is there a way to perform simulation to assess this?
- There does not appear to be any discussion of existing literature on competing methods. Using gene co-expression patterns to infer trait-relevant tissues is not novel in the field, such as CoCoNet. There are also many tools/papers (such as LDSC-SEG, deTS and RolyPoly) using cell/tissue specific expressed genes to infer trait-relevant tissues. Please compare PhenoPLIER results with the results of these tools on the same dataset (eg. 4,000 gene-trait associations used in this study).
- The methodology seems incremental. Based on Figure 1b, PhenoPLIER, the method that the authors propose, simply multiplies Z transpose with M, which are outputs from two previously existing methods, MultiPLIER and PhenomeXcan (or LINCS), respectively.

- Why was a decision tree classifier (CART) chosen instead of other tree-based methods such as random forest? The latter usually has better performance and generalizability.
- Precision-recall curve decreases to 0.75 seems strange, is the scale correct?
- In Figure 1a, the TWAS label should be near the arrows point from genes (middle) to the diseases (top). From the genotype to transcripts would be eQTL. Also in Figure 1a, why are the associations represented by directed arrows? The associations should not be undirected.
- Figure 1b: the dimension of Z transpose, which is latent variable (LV) by gene, is not properly represented in the figure. It should be a wide (rather than narrow) matrix. Also, at the top of the plot, the \sim or tilde symbol should be an \approx symbol instead (based on my understanding).
- The method description is vague. The correspondence between specific subsections of the results section and the methods section is unclear. The section "LV-trait associations via gene-property analysis" is hardly comprehensible.

Response to reviewers

Reviewers' comments are in black, and authors' responses start with "Authors' response" and use blue as the font color.

Reviewer #1

The manuscript by Pividori et al presents a set of integrative analyses that merge together gene-trait associations (from TWAS) with gene perturbation data to gain insights into which groups of genes are impacting which group of traits. The key insight of the approach is that gene modules (i.e. group of genes that share co-expression patterns) are more informative than modeling single gene entities. The manuscript is extremely densely written with many different analyses that are hard to follow, insufficient motivation for proposed analyses, and very few (if any) comparisons to alternate approaches to guide the parameter selection. Thus, I am moderately enthusiastic for the proposed manuscript.

I hope comments below are useful for the authors:

Authors' response: We want to thank the reviewer for this constructive feedback. We found these comments very useful in restructuring the manuscript to more clearly present the key insight of the approach and better guide the reader through the manuscript and the different analyses performed in support of the key insight.

Regarding motivation, PhenoPLIER is an omnigenic approach that provides a gene module perspective to genetic studies. To our knowledge, PhenoPLIER is a computational framework for data integration that answers key research questions of great importance today, from prioritization of causal genes missed by standard genetic studies to providing a mechanistic interpretation via gene modules expressed in specific cell types. We now more fully summarize the motivation in the first section of Results, where we list the questions that PhenoPLIER could help to answer.

We significantly improved the introduction and Figure 1 to more clearly motivate and present our work. We examined the effectiveness of PhenoPLIER with discovery and replication cohorts (the UK Biobank and eMERGE), where validation of our integrative approaches can be grouped into three primary analyses:

1. Gene module-trait associations to infer how groups of functionally-related genes influence complex traits (regression model).
2. Detection of shared and distinct transcriptomic properties among traits (clustering).
3. Prediction of how pharmacological perturbations affect genes' activity to exert their effects (drug repurposing).

We believe these changes significantly improved the readability of the manuscript and more clearly delineate primary findings from supporting evidence.

The reviewer also commented that there are *“very few (if any) comparisons to alternate approaches to guide the parameter selection.”* As we detail below, in our cluster analyses, we perform parameter selection using consensus clustering and well-known and established algorithms.

1. Core to the approach is the concept of gene modules (latent variables, LD) that are reconstructed within a matrix factorization approach (MultPLIER). How are the modules (LVs) robust to false network discoveries? E.g. recent works (Parsana et al Genome Biol 2019) note that regular technical/biological artifacts confound

common gene expression modules with PC correction being able to reduce false discoveries. Is a PC correction performed here? Are the LV robust to different reconstruction methods?

Authors' response: These are important aspects of the methodology proposed in our work. We used latent variables (LVs) obtained from the MultiPLIER study, which used the Pathway-level information extractor (PLIER), a matrix factorization method (Mao, Nature Methods, 2019).

The first question raised by the reviewer is about LV robustness to technical/biological artifacts. PLIER allows specifying prior knowledge for the training process in the form of a binary matrix with genes and their membership to known pathways. This prior knowledge guides the factorization method by learning LVs that align well with known mechanisms (pathways). At the same time, potential technical factors are segregated into LVs that do not associate with prior information (more details in Mao, Nature Methods, 2019). LVs that do not align well with pathways might represent either novel transcriptional mechanisms or technical artifacts. In our work, we focused our primary analyses on LVs that were aligned with prior information, while also providing the results across the entire set of LVs in the supplementary material. This is the way PLIER deals with technical factors. The method suggested by the reviewer (Parsana, Genome Biol, 2019) is only another way (out of many) to deal with confounding, which also has problems such as the removal of real signals, as noted by those authors. However, more importantly, these correction techniques could be detrimental in larger datasets like recount2 (used by MultiPLIER). For example, in recent work (Lee, Gigascience, 2020), the authors found that datasets with several sources of variation, such as recount2 (used in that study and ours), correction hampers performance by removing real signal instead of technical artifacts, concluding that "*when many experiments are combined, statistical correction reduces our ability to extract underlying patterns.*" It is also worth noting that for PhenoPLIER to identify an association, the LV would need to be associated for non-biological reasons in TWAS as well as gene expression data. In summary: 1) the models we used in our work already provide a way to segregate technical factors into LVs that are differentiated from those well-aligned with known pathways, 2) given the size of the expression compendium (recount2), a correction technique is discouraged, and 3) that issues would only arise if technical artifacts were consistent across gene expression and association data.

The second question is about the robustness to different reconstruction methods. The challenge of learning LVs from expression data is not the main point of our study. In our study, we integrate LVs with genetic studies. MultiPLIER / PLIER is only one of several approaches to extract patterns from expression data. These approaches use different assumptions to model the data and extract patterns, which could still be useful when we integrate them with genetic studies, even if they do not replicate across different reconstruction methods. For example, we could replace the LVs from MultiPLIER with those extracted by GenomicSuperSignature (Oh, Nature Communications, 2022), a recent approach that uses different assumptions to learn these patterns. Rather than dealing with the extraction of LVs from expression data, in our study, we provide a methodology to assess whether LVs are useful to:

1. better understand disease-relevant mechanisms expressed in specific contexts (LV-trait association with a regression model),
2. detect shared mechanisms across a group of diseases (clustering), and
3. their potential relationship with known drugs (drug repurposing).

If the LV models provide information about specific contexts/cell types/tissues (like MultiPLIER), our approach could also aid in the interpretation of results, as we show in our study.

2. An LV-trait association p-value is proposed to connect modules to traits. Is the p-value calibrated? I could not follow what is the null model rejected here; is that all genes in a module are not associated to the trait? At

minimum the authors should present some null simulations to showcase that their proposed approach is not subjected to false positive rates.

Authors' response: This is a very important question that helped us to improve our regression model and successfully verify that the p-values are properly calibrated. In Methods, we now provide more details about our LV-based regression approach. We also added Supplementary Note 1 with the null simulations, where we used data from 1000 Genomes to simulate a thousand (1000) normally-distributed random phenotypes. On these randomly simulated phenotypes, we ran GWAS, harmonization and imputation procedures (see Methods), TWAS, and finally, our LV-based regression model. We verified that the mean type I error of our approach is acceptable and smaller than a naive linear regression without accounting for gene-gene correlations. We also show that LVs with adjacent genes at the top (which tend to have correlated predicted expression in TWAS) are corrected in our approach. We ran our LV-based regression model on all 4,091 traits in PhenomeXcan (discovery cohort) and 301 phecodes in eMERGE (replication) across all 987 LVs and observed no signs of inflation (Supplementary Figure 14). We note that, as we describe below, the gene-gene correlation matrix is now estimated using GTEx v8, which is the most appropriate reference panel since it has to match the one used in TWAS (this is also discussed below).

3. The TWAS results integrated here are coming from a single TWAS method applied to UKBiobank data; first, the authors could investigate the robustness of their results to the PrediXscan method. But most importantly, given that the GWAS data utilize the same set of individuals will induce covariances in the TWAS statistics (due to shared environments) that will create spurious covariances in any linear combination of TWAS z-scores. The authors could investigate how these correlation may bias their approach.

Authors' response: We believe that analyzing the robustness of our approach to different TWAS approaches is beyond the scope of our study, and it would not be beneficial in evaluating our methods. TWAS approaches that are "compatible" with PhenoPLIER (those that provide a p-value for a gene and a trait like S-PrediXcan / Fusion (Gusev, Nature Genetics, 2016) or S-MultiXcan / UTMOST (Hu, Nature Genetics, 2019)) are generally similar (Weinberg, Nature Genetics, 2019).

As the reviewer says, the GWAS data we use are more important. However, the idea that GWAS results are based on the same set of individuals is incorrect, which we have now clarified. First, we used a discovery (PhenomeXcan) and a replication (eMERGE) cohort. PhenomeXcan provides GWAS/TWAS results for 4,091 traits, and eMERGE (replication) provides independent GWAS/TWAS results for 309 phecodes. In PhenomeXcan, most of the traits are from the UK Biobank, but PhenomeXcan also provides 42 well-powered GWAS/TWAS conducted in different sets of individuals. Several LV-trait pairs significantly associated in PhenomeXcan were also clearly replicated in eMERGE. Among these, LV246 (presented in section "LVs link genes that alter lipid accumulation with relevant traits and tissues") is strongly associated with lipids, high cholesterol, cholesterol-lowering medication and other cardiovascular-related traits in PhenomeXcan (Supplementary Table 8). As expected, most of these significant associations in PhenomeXcan are from the UK Biobank. However, the top two traits (triglycerides and LDL cholesterol) were profiled by (Kettunen, Nature Communications 2016) using a different set of individuals. In addition, LV246 is strongly associated with hypercholesterolemia (phecode 272.11) in eMERGE (Supplementary Table 9).

This pattern is also the same in our cluster analyses. In section "LVs reveal trait clusters with shared transcriptomic properties," many of the LVs driving different groups of traits are associated with the same or highly-related traits/diseases from different cohorts. An example is LV844, which separates the cluster of autoimmune diseases from the rest very well. This cluster contains GWAS from the UK Biobank (such as celiac disease or hyperthyroidism) but also from other cohorts, such as rheumatoid arthritis (Okada, Nature, 2014) or systemic lupus erythematosus (Bentham, Nature Genetics, 2015).

We clarified these details about the TWAS methods and GWAS data in the manuscript. Regarding TWAS methods, we now improved (In the Methods section "The PrediXcan family of methods for gene-based associations") the description of the TWAS approaches employed in our study (S-PrediXcan and S-MultiXcan) and how they were used across our analyses. For example, S-MultiXcan results were used for the LV-trait regression framework and cluster analyses, whereas S-PrediXcan was used for the drug-repurposing approach (since it provides the direction of effect). Regarding the GWAS/TWAS data, we extended our Methods section (including subsection "TWAS resources") to explain better the cohorts used. We used PhenomeXcan as the discovery cohort and eMERGE as the replication cohort. We also improved Figure 1, which now provides a better schematic of our approach. We also added in Supplementary File 1 a list of all 4,091 traits in PhenomeXcan, which includes the study they come from (such as UK Biobank, IGAP, among others). The list of traits/phcodes in eMERGE is provided in (Li, medRxiv, 2021, <https://doi.org/10.1101/2021.10.21.21265225>).

4. What is the impact of using out-of-sample LD from 1000 Genomes instead of in-sample LD in equation 3? Recent works that prioritize TWAS z-scores to find causal genes note that such reference LD may bias inference (Mancuso et al Nat Genet 2019). The authors could easily use the UKBiobank LD to check robustness in their results.

Authors' response: This important point allowed us to improve our LV-based regression model. We significantly extended the Methods sections with details about the TWAS approaches (subsection "The PrediXcan family of methods for gene-based associations") and our LV-based regression model (subsection "Regression model for LV-trait associations.") As explained there, the most appropriate reference panel in our case is GTEx v8 (not 1000 Genomes as we used previously) since this is the reference panel used by the TWAS methods to compute all the statistics in UK Biobank, eMERGE and any cohort. For instance, in S-PrediXcan (now Equation (2)), GTEx v8 is always used (regardless of the GWAS cohort) to compute genotype variances and covariances. And the same happens for S-MultiXcan (now Equation (4)) to compute the correlation between the predicted expression across tissues for a gene. In other words, the TWAS methods we used are all summary-based and always rely on GTEx v8 for LD information. Therefore, in our regression approach (a generalized least squares model) we need to match the LD reference panel used by the TWAS methods to properly account for correlations between genes' association statistics.

In our null simulations for the regression model (Supplementary Note 1), we simulated a real application of TWAS and our LV-based regression model, where the GWAS cohort (in this case, 1000 Genomes) does not match the LD reference panel used by TWAS and our LV-based regression model (GTEx v8). These results show that the p-values are approximately well calibrated (Supplementary Figure 8), and scenarios with adjacent and highly correlated genes at the top of some LVs are properly accounted for (Supplementary Figures 9-12).

5. In the clustering analysis, 5 clustering approaches are applied to the same data and a consensus is taken together. Same input data and similarities in the clustering algorithms will induce covariances in the clustering results; how are such expected covariances taken into account in the ensemble method? Second, the proposed ensemble approach may be overfitting the data; I suggest to do cross-validation outside(prior) to the application of 5 clustering approaches (in addition to the cross-validation within each clustering approach to select their best parameters).

Authors' response: This is an interesting point. We have performed new analyses based on the reviewer's suggestions to examine the concerns raised. We improved the section "Consensus clustering of traits" (in

Methods). We also added Supplementary Note 2 with simulations on two different scenarios with no structure in the data and confirmed that our results reflect a real structure at different resolutions.

Regarding the first point, we would like to clarify that the consensus clustering framework is an established technique to robustly detect structure in data (we did not invent a new clustering method). Consensus clustering has been suggested to address many of the most common pitfalls when running cluster analysis on complex biological data (Ronan, *Science Signaling*, 2016), including the reviewer's concerns about biases, for example, due to the selection of a single clustering algorithm. Consensus clustering has two phases: 1) ensemble generation and 2) ensemble combination (consensus). In our study, and as we show in Figure 5 and explain in Methods (subsection "Consensus clustering of traits"), consensus clustering was applied to different transformations of the input data (3,752 traits and 987 LVs): each of the five clustering methods was applied to 1) the raw input data, 2) the PCA transformation with 50 PCs and 3) the UMAP transformation with 50 components. Then, as described in Methods, each clustering algorithm was applied with different parameters, such as the number of clusters or parameters ϵ and $minPts$ in DBSCAN. The five clustering algorithms are very different, with entirely different assumptions on the data structure: k-means, for instance, can find hyperspherical clusters, whereas spectral clustering can find more complex structures by using the spectrum of the similarity matrix from the input data. DBSCAN is a density-based approach with a very different set of parameters and assumptions, and hierarchical clustering was applied with three different linkage criteria (important parameters of the method that can lead to different results). The reason for generating different data versions and using different algorithms, as in any application of consensus clustering, was to generate a highly diverse ensemble. Diversity has long been recognized as a critical element in these ensemble-based approaches (Hadjitodorov, *Information Fusion*, 2006; Lam-On, *Pattern Anal. Mach. Intell.*, 2011; Pividori, *Information Sciences*, 2016). After the ensemble is generated, the consensus approach creates a squared co-association matrix (with traits in rows and columns) representing the number of times each trait pair was clustered together. Then, a clustering method (spectral clustering in our case) is applied to the co-association matrix to derive the final consensus solution. Briefly, if the solutions in the ensemble, generated from different points of view, agree on grouping together a pair of traits in the consensus phase, that is evidence that the link between the pair of traits is strong. To illustrate this, we provide below some plots showing the co-association matrix for a pair of clusters in our results, where numbers in each cell represent the fraction a pair of traits in that cluster was grouped together in the ensemble (generated with different data versions and different algorithms):

Keratometry cluster (Cluster number 10 in Figure 6)

Heel bone densitometry cluster (Cluster number 15 in Figure 6)

For example, traits with codes 4124 and 3148 in the second plot (Heel bone densitometry cluster) have a value of 0.89, meaning that 89% of the ensemble partitions have grouped this trait pair together. These co-association values (by definition, from 0 to 1) are high; the entire co-association matrix has a median of 0.47 and 75th and 99th percentiles of 0.50 and 0.69, respectively.

Regarding the concern about clustering models overfitting the data, we applied several known techniques for parameter selection at the final consensus phase (see Methods). In the ensemble generation phase, however, we did not pick a single number of clusters since we need to generate diversity at this step. However, we provide some plots below to reply to the reviewer’s comments. At the ensemble generation step, for each clustering algorithm, we used a range of different parameters (such as the number of clusters or the e and $minPts$ parameters in DBSCAN) and computed several internal quality measures, including the Calinski and Harabasz score and the consensus index (Vinh, Journal of Machine Learning Research, 2010) to confirm the solutions generated had a minimum level of quality (for example, that not all points were detected as noisy by DBSCAN). For example, we show below the consensus index values (representing the stability of solutions) for k-means on the raw data and PCA transformation for k values from 2 to 60, where it can be seen that k-means is more stable on lower values of k .

Consensus index (measured as average adjusted Rand-index/ARI) for k-means on raw data and k values from 2 to 60.

Consensus index (measured as average adjusted Rand-index/ARI) for k-means on the PCA transformation of the data and k values from 2 to 60.

In addition to showing that k-means has the same trend in both data versions (i.e., more stable solutions on lower k values), the plots also suggest that the different data versions were useful as a source of perturbation for this algorithm. The same pattern was found for the other algorithms as well. At the ensemble generation step, we selected all partitions from 2 to 60 to improve diversity for k-means, spectral clustering and Gaussian mixture models. A similar approach was used for other algorithms without a k parameter, such as DBSCAN.

At the consensus phase (where we generated the final solutions shown in Figure 6), we performed parameter selection (number of clusters) using a data-driven method (explained below). We plotted the results in a clustering tree (Figure 6) because selecting a single partition would ignore the complexity of associations, where we do not expect a clean separation at one specific resolution. For this, we combined the highly-diverse ensemble into final consensus solutions using different numbers of clusters (Methods). To select the final number of clusters at this stage, we used a known technique based on the average agreement of the final consensus partition with the ensemble (Strehl, Journal of Machine Learning Research 2002; Ghosh, NSF NGDM, 2002). This led to 15 final clustering solutions we plotted using a clustering tree (Zappia, Giga Science, 2018) in Figure 6. This reveals how traits move across different resolutions using the clustering tree. Zappia et al. (GigaScience, 2018) provide clustering trees under control conditions that lack structure (Figure 2 from their manuscript, reproduced below):

Our clustering tree (Figure 6, this manuscript), does not appear to follow the negative control pattern suggesting that there exist different and valid solutions in the data with varying resolutions (number of clusters), forming branches of trait clusters that resemble known biology (such as cardiovascular or autoimmune diseases from GWAS obtained from different cohorts). As another validation step, we detected which LVs were driving these clusters using a decision tree classifier (Methods). Using our LV-based regression framework, we found that these LVs were significantly associated with the same or very similar traits in our replication set (eMERGE). For example, LV57 was significantly associated with deep venous thrombosis and hypothyroidism in the discovery and replication cohorts (Supplementary Tables 26 and 27); or LV136 with coronary atherosclerosis (Supplementary Tables 32 and 33).

Finally, the reviewer suggested performing cross-validation for parameter selection. As explained before, we performed parameter selection using different approaches but not cross-validation for a set of specific reasons. First, cross-validation approaches for clustering, which are similar to stability-based analyses, assume a single “true” structure in the data. This is a very strong assumption that is usually unrealistic in complex and high-dimensional biological data such as ours, where multiple true structures could provide different insights (Ronan, Science Signaling, 2016). In contrast with supervised techniques, clustering is an ill-posed problem where different and equally valid solutions may exist. Cross-validation applied to cluster analyses, such as the “prediction strength” (Tibshirani, J. Comput. Graph. Stat. 2005), has been proposed to select the number of

clusters. The procedure consists of splitting the data into training and test sets, applying the clustering algorithm to both, then using the model learned from the training set to predict labels in the test set and compare them. Although valid, this approach has several drawbacks. For instance, in the previously referenced article, the authors acknowledge that the technique works for simple cluster shapes but not for elongated ones, limiting its applicability to data with clear, simple, and often unrealistic cluster structures in real data. We expect this model is too simple for association data, with complex genetic relationships between traits. However, more importantly, the "prediction strength" method assumes that test samples can be predicted using the centroids learned from the training data. This dependence on centroid-based algorithms, such as k-means, dramatically limits its applicability and makes it infeasible for other algorithms, such as a density-based method like DBSCAN or a consensus clustering framework (the one we used) where centroids are meaningless. As an additional analysis, we performed simulations with the null hypothesis of no structure in the data (Supplementary Note 2). As shown in Supplementary Figure 15 and copied/pasted below, we found that solutions on PhenomeXcan (labeled as "real data") significantly deviate from the expected ones under the assumption of no structure in the data (more details in Supplementary Note 2). The consensus clustering approach fails to detect any meaningful partition from the ensemble generated using the scenarios with no structure in the data (Null #1 and Null #2), where the agreement of the consensus partitions at all k values is very close to zero.

Supplementary Figure 15. Agreement of consensus partitions with the ensemble. A real and two simulated scenarios with no data structure are shown. For each scenario, one final consensus partition was derived for each k from 2 to 60 (x-axis) following our clustering pipeline (Methods). For each partition, the agreement with the corresponding ensemble was computed using the adjusted RandIndex or ARI (y-axis). For the real data scenario, partitions with an agreement above the 75th percentile (dashed line) were selected for follow-up analyses in the main text (shown in Figure 6 with the clustering tree).

6. I suggest the authors separate the discussion section to improve readability.

Authors' response: We agree that this was not well organized in the first submission. We moved all the discussion parts in Results into the Discussion section.

Reviewer #2

In this work, the authors derived factorized co-expression gene modules weights from large-scale expression compendium by using previous MultiPLIER method. And they proposed a framework PhenoPLIER to test these co-expressed gene-trait associations and co-expressed gene-drug signature associations by utilizing PhenomeXcan and LINCS L1000 data, respectively. The authors identified a few significant associations and used CRISPR screen to validate one of their findings in lipid trait. Overall, it is a systematic design. However, this work heavily relies on their previous matrix factorization-based method MultiPLIER. The majority of the results are using the same analytical strategy to interpret the significant associations in latent variables (LVs) projections. The major innovation parts are the methodology of adding PhenomeXcan results and drug signature to the LVs and subsequent clustering analysis of the M^A and decision tree on traits. So I have a few questions in terms of these two methods.

Authors' response: We want to thank the reviewer for the positive feedback and valuable comments. We want to clarify that our approach does not rely heavily on MultiPLIER but only on any model that learns a latent representation from gene expression data. As we mention in the Discussion section, we could use, for instance, GenomicSuperSignature (Oh, Nature Communications, 2022), a recently published approach that uses different assumptions to learn these gene co-expression patterns.

For the PhenoPLIER model:

1. Many regulatory SNPs and pleiotropic effect have population-specificity. TWAS and fastENLOC results are also population sensitive. However, the gene expression compendium might not have the corresponding population information. Thus, how PhenoPLIER model could address the population background difference from gene expression and GWAS trait?

Authors' response: This is an important question, and it is linked to future work. As the reviewer notes, all of our TWAS results are from European ancestry, and recount2, the gene expression compendium used by MultiPLIER, has several sources of heterogeneity: different tissues, cell types, samples under different stimuli, disease status, among many others, potentially including different population ancestries, as the reviewer mentioned. Below, we provide two answers based on whether we consider that this "correction" for population background is needed or not.

First, we used a latent gene expression representation that aligns latent variables (LVs or gene modules) to prior knowledge (pathways). This is similar to other techniques that compute an association between known pathways and gene-trait scores, such as MAGMA (de Leeuw, PLOS Computational Biology, 2015). And we primarily focused on well-aligned LVs, while providing the complete set of results across all LVs in our Github repository. Most of the LVs are expressed in specific cell types or tissues (such as LV246 in adipose tissue or LV136 in the pancreas and liver), or under different stimuli (such as LV116, shown in Figures 4 and S17). As the reviewer mentioned, we do not have metadata in recount2 to assess whether the LV captures a population-specific pattern. However, we observe that LVs are useful for finding expected groups of traits in our cluster analyses and better predicting known drug-disease links. Several gene modules like LV246 (aligned with lipid metabolism pathways) are expressed in relevant tissues/cell types and associated with highly-related traits (lipids measures, high cholesterol, among others). We found that these associations also replicated in the European-specific eMERGE data. In summary, if a match between TWAS and LV population ancestries is a concern, our pipeline provides the tools to ensure this consistency through LVs aligned with prior knowledge (pathways) and a replication cohort with European-specific TWAS results.

The second aspect of our answer is whether this "correction" of population-specific LV patterns is really necessary. In this case, the assumption is that LVs that align well with known pathways represent more conserved mechanisms than single SNPs or genes across different population ancestries. What would it mean if LV246, for example, is mainly expressed in adipose tissue from non-European samples from the expression data? In this case, the hypothesis that LV246 represents a genetic feature that is more conserved across different ancestries than single SNPs or genes is supported by the fact that 1) this LV is associated with the same traits in both the discovery and replication cohorts (both from European ancestry), and 2) its genes are expressed in highly relevant cell types (adipose tissue). In the future, extending our TWAS results to non-European populations, or improving gene expression metadata, would allow us to verify this hypothesis. Given the scope of the current manuscript, we expect this would be an exciting path for future research.

2. Why the model only consider the magnitude of S-MultiXcan without considering the effect direction when combining the TWAS effect to LV? Can the author give an explanation for choosing multiplicative model as the combining strategy of the weights instead of other models, such as additive model?

Authors' response: This is another important question, and it was useful to clarify further the TWAS methods used in our study. In subsections "The PrediXcan family of methods for gene-based associations" and "TWAS resources" (Methods), we improved our explanation of the TWAS methods (S-PrediXcan and S-MultiXcan) and the TWAS resources used (PhenomeXcan as discovery cohort, and eMERGE as replication cohort).

In our study, we used both S-PrediXcan (with the direction of effect) and S-MultiXcan (without the direction of effect). S-PrediXcan provides a p-value of association for a gene-tissue-trait triplet and its direction of effect. S-PrediXcan was used in our LV-based drug-repurposing framework because we need to know whether a higher or lower predicted gene expression is associated with disease risk. Combined with drug-induced transcriptional profiles using positive or negative z-scores (meaning that the drug increases or decreases the expression of the gene), this allows us to compute a score for a drug-disease pair based on the procedure published in (So, Nature Neuroscience, 2017). Therefore, we do use the direction of effect in the LV-based drug-repurposing analyses.

On the other hand, S-MultiXcan computes the joint association between a gene's predicted expression in all tissues and a trait (Equation (3)). Since S-MultiXcan is more powerful than S-PrediXcan in detecting gene-trait associations (Barbeira, PLOS Genetics, 2019), we used it in our LV-based regression framework and cluster analyses. In these analyses, we do not need the direction of effect but only the p-value of association (which S-MultiXcan yields). This is because, in the context of gene modules or LVs, we are not interested in the direction of effect of single genes but in their association as a module or group with a trait. We significantly improved the Methods section by more clearly describing the TWAS methods, resources, LV-based regression framework, cluster analyses and LV-based drug-repurposing approach.

Finally, the reason to multiply the LV weights by the gene-trait p-values (converted to z-scores) is that we want to project these gene-trait associations into the latent space for a joint analysis of traits (in the case of cluster analysis) or the prediction of drug-disease pairs (drug-repurposing). By multiplying LV weights with gene-trait z-scores, we add evidence from a module on whether its top genes are important for that trait or drug. We note that this projection is not related to an additive model as in a GWAS study.

3. TWAS analysis either from S-PrediXcan or S-MultiXcan could only infer the association between GWAS traits and GTEx available tissues. What is the rationale by adding TWAS effect to the LVs of unrelated tissues or cell types factorized from gene expression compendium?

Authors' response: This is a critical question. As the reviewer says, TWAS results rely on available tissues in GTEx v8, whereas LVs were extracted from recount2 (without GTEx samples). GTEx v8 covers a wide range of normal human samples, with more than 30 solid-organ tissues, different brain subregions, whole blood, and cell lines from blood and skin samples. On the other hand, recount2 provides a highly diverse set of contexts, including different tissues, cell types, stimuli, and diseases, among many others. Many of these contexts in recount2 will be unrelated to GTEx tissues, but many others will relate more or less directly. An example of a more direct relationship is module LV246 (Figure 2a and Supplementary Table 7), strongly aligned with lipids metabolism pathways and whose genes are clearly expressed in adipose tissue, which is one of the 49 tissues in GTEx v8. A more indirect example is module LV603 (Figure 1c and Supplementary Table 3), whose genes are clearly expressed in neutrophils and other white blood cells, which is expected since this module is strongly aligned with neutrophil-related pathways. Although GTEx does not provide expression panels on neutrophils, this type of granulocyte is generally the most abundant leukocyte in human peripheral blood, a tissue available in GTEx v8. In summary, we expect that a fraction of the diverse set of contexts in recount2 will match, with different degrees, to a subset of the range of tissues provided in GTEx v8. In our analyses, we expect no associations between entirely unrelated contexts (note that we have now added null simulations about this, detailed below).

On the other hand, we always consider TWAS associations as evidence of a link between a gene and a trait, regardless of the tissue model in GTEx. We note that in our analyses, we never consider tissue-specificity from TWAS (S-PrediXcan results). We always consider TWAS associations (from S-PrediXcan and S-MultiXcan) across *all* available tissues. This is the approach taken by S-MultiXcan, which computes the joint association between *all* tissues available for a gene and a trait. And when using S-PrediXcan in our drug-repurposing framework, we take the maximum prediction score across *all* tissues as well. By doing this, we gather all the evidence we have from GWAS and GTEx that a gene is associated with a trait regardless of context, which is later integrated with LVs or groups of functionally-related genes.

In summary, the rationale for integrating TWAS association data and LVs from large expression datasets is the hypothesis that *real* gene-trait associations should match *real* groups of functionally-related genes (LVs or gene modules) expressed in specific contexts. In more technical terms, if an LV represents a real gene module expressed in a certain context, then genes that are causally linked to a trait should tend to belong more strongly to the LV (i.e., have large weights). Our results, based on discovery and replication cohorts, suggest that this approach (PhenoPLIER) works across different analyses and is able to effectively integrate different data modalities. We have sought to clarify our writing, and we have added null simulations for several of our analyses that should aid in interpretation:

- In Supplementary Note 1, we added simulations for our LV-based regression framework and found that its p-values are approximately well-calibrated under the null hypothesis of no associations.
- In Supplementary Note 2, we added simulations where we show that our clustering results significantly deviate from a scenario of no structure in the data.

4. In the PhenoPLIER model, the S-PrediXcan results were projected to M^A is not well-described, whether they has direction or not. And how it will be different from the S-MultiXcan model.

Authors' response: We thank the reviewer for this comment. We agree that this part was not clearly described, and we have clarified these aspects in the manuscript (subsection "LV-based drug repurposing approach" in Methods). We also added the subsection "The PrediXcan family of methods for gene-based associations" in Methods, with more details about S-PrediXcan and S-MultiXcan, and how they were used. In both the single-gene (So, Nature Neuroscience, 2017) and our equivalent LV-based drug-repurposing frameworks, we used the signed z-scores (with the direction of effect) from S-PrediXcan. Here we need the

direction of effect to infer whether a higher or lower predicted gene expression is associated with disease risk. Then, this was combined with equivalent information for a drug from LINCS L1000 (whether a drug increases or decreases the gene expression) to compute a drug-disease link.

On the other hand, S-MultiXcan was used in our LV-based regression framework and cluster analyses of traits since S-MultiXcan is more powerful than S-PrediXcan in detecting an association between a gene and a trait. S-MultiXcan, however, does not provide the direction of effect because it combines all available tissues for a gene, which makes it unsuitable for the drug-repurposing framework presented in (So, Nature Neuroscience, 2017). However, this increase in power is more important for the regression and clustering tasks.

For the Cluster analysis on traits method:

1. The method of using 5 clustering algorithms to measure the consensus clustering seem inspiring and could be expanded to other scenarios for the readers. Can the authors comments on the methods are the state-of-the-art method or not?

Authors' response: We appreciate these comments from the reviewer on our cluster analyses. Indeed, consensus clustering or cluster ensembles are a state-of-the-art set of methods widely used in different areas. It has been shown that these ensemble-based approaches outperform individual clustering algorithms (Strehl, J. Mach. Learn. Res., 2002; Fred, Pattern Anal. Mach. Intell., 2005; Lam-On, Pattern Anal. Mach. Intell., 2011) by providing more accurate and robust solutions. These techniques also simplify a clustering task for a user since he or she does not need to previously select a single clustering algorithm (which could bias results) or spend lots of time setting up a method for a particular scenario that might require very specific parameters. A fundamental property for consensus clustering methods to work, however, is the diversity of the ensemble (Pividori, Information Sciences, 2016). That is why we generated different data versions (raw, PCA, UMAP) and used different algorithms (under different assumptions on cluster shapes) with different sets of parameters (like the number of clusters in k-means).

For biological datasets, consensus clustering has been suggested as a solution to address many of the most common pitfalls (Ronan, Science Signaling, 2016) in the biology domain. In that article, the authors present consensus clustering as *"an easy-to-implement method that enables the exploration of multiple clustering solutions and improves robustness of clustering solutions."*

2. I wonder how robust these decision trees will be when the latent variables pool changes along with the accumulation of the gene expression compendium. On the other hand, I wonder what is the composition of the 3,752 traits? And how could they be clustered in only 29 groups with biological meaningful interpretation?

Authors' response: These are important and interesting questions. Regarding the first point, the LVs in MultiPLIER were extracted using pathway information as prior knowledge. Methods to reduce dimensionality largely learn an arbitrary rotation of the data, so we would not necessarily expect the same axes. However, prior knowledge in PLIER constrains the rotation in ways that would likely retain consistency. If the same approach is used with larger datasets such as recount3 (Wilks, Genome Biology, 2021) or ARCHS4 (Lachmann, Nature Communications, 2018), using prior knowledge would imply some stability in the learned models since this is fixed information that is used in the training of the matrix factorization approach. Therefore, we would expect, for example, an LV equal or very similar to LV246 (aligned with lipids metabolism) to be extracted again in larger compendia if prior knowledge is used.

Regarding the composition of the traits in PhenomeXcan (discovery cohort), we have added Supplementary File 1, which includes basic information about the traits, such as UK Biobank IDs (allowing mapping to

categories of traits), description, and sample size. More than four thousand traits in PhenomeXcan are from the UK Biobank (from Neale's lab Rapid GWAS Project, round 1, <http://www.nealelab.is/uk-biobank>), so most of the trait categories present in that cohort were used, including a wide range of diseases, sociodemographics, lifestyle, early life factors, among others. For eMERGE (replication cohort), we refer to (Li, medRxiv, 2021, <https://doi.org/10.1101/2021.10.21.21265225>) for details about traits.

Finally, cluster analysis of this data is challenging since we expect several complex traits to be related and hard to categorize, as we know from pleiotropy analyses (Watanabe, Nature Genetics, 2019). Given the challenging task and the large number of traits in PhenomeXcan, we believe that our results are acceptable, with clusters representing a set of known disease domains that cover many of the categories we have in the data. Our simulations and results show that we could potentially extract more clusters of traits using the same approach (Supplementary Figure 15). However, we decided to focus on those with the most significant agreement with the ensemble to prioritize those groups that are more stable. In the Discussion section, we describe some limitations of our cluster analyses, including the type of algorithm used (hard-partitioning, where one trait belongs exclusively to one cluster) and the distance metric used to compare a pair of traits (which takes into account all features/LVs). Although the cluster analyses using this design proved helpful, these two properties (hard partitioning and distance metric using all features/LVs) pose several limitations. First, one trait could strongly belong to different clusters instead of exclusively being part of a single group. Second, although two traits could be globally similar (as we capture in our analyses), it could also be possible that they are similarly related when considering only a reduced subset of LVs. To address these points, we would need to apply a fuzzy clustering algorithm, a biclustering approach or a matrix factorization technique.

Reviewer #3

In this manuscript, Pividoret al. developed PhenoPLIER, a method that integrates TWAS datasets or drug profile data through latent variables (gene modules) and claims that using gene modules improves interpretability in the results.

Using MultiPLIER, latent variables are first learned from various bulk RNA-seq data (recount2) represented by gene modules and reduces dimensionality to 987 latent variables. Then, it uses TWAS data from PhenomeXcan and transcriptional profiles of drug data from LINCS L1000 and projects these into the same low-dimensional latent variables. Several examples were included: 1) identified diseases associated with gene modules such as hypothyroidism with T cells. 2) performed a CRISPR screen that showed more robust trait associations with gene modules than single genes. 3) drug disease prediction was better using gene modules than single genes. 4) clustering the projected TWAS data then training a decision tree classifier showed insights on transcriptional pathways.

Authors' response: We want to thank the reviewer for the thoughtful comments that helped us to improve our manuscript significantly. First of all, based on these and the other reviewers' comments, we significantly improved our manuscript in the following areas:

- The Introduction was simplified and improved, where we better present our approach and focus on gene modules integration with GWAS/TWAS, which is the primary goal of PhenoPLIER. We noticed that the introduction was confusing about inferring tissue-specific effects, so we changed it to reflect better the work in this manuscript, which is about gene modules and thus only indirectly about tissue specificity (more details below).

- Figure 1 was improved to show the main components of our proposal: 1) an LV-based regression framework, 2) cluster analyses of traits in the LV space, and 3) an LV-based drug-repurposing framework.
- Figure 1 now clearly shows that we are using PhenomeXcan as a discovery cohort and eMERGE as a replication cohort.
- The Methods section was also significantly expanded, with more details for all of our analyses and methods.
- We added null simulations for our LV-based regression framework (Supplementary Note 1), which allowed us to verify that the model is approximately well-calibrated, with acceptable mean type I errors.
- We added simulations for our clustering analyses (Supplementary Note 2), which allowed us to verify that the structure learned from the data significantly deviates from a scenario of no structure.
- As the reviewer pointed out, we improved the clarity and description of our analyses across the manuscript.

However, using gene co-expression patterns to infer trait-relevant tissues is not novel in the field, such as CoCoNet (Leveraging gene co-expression patterns to infer trait-relevant tissues in genome-wide association studies, PLoS Genet. 2020). There are also many tools/papers (such as LDSC-SEG, deTS and RolyPoly, as below) using cell/tissue specific expressed genes to infer trait-relevant tissues. Comparing with these tools, I do not think PhenoPLIER has big improvement, maybe even worse. In the manuscript, the authors only compare PhenoPLIER results with TWAS results (it's a little strange because TWAS results are the input data of PhenoPLIER) but not with these similar tools. On the other aspect, PhenoPLIER can expand TWAS genes with co-expression network, as example, the authors found that lipids-related genes DGAT2 and ACACA cannot be identified TWAS but PhenoPLIER. However, the authors did not analyze how many false positive genes (I guess a lot) can also be included by co-expression network. I am sure I can also find many functional trait-related genes by simply expanding TWAS risk genes with protein-protein interaction network or pathways. So, the DGAT2/ACACA example cannot convince me. BTW, the manuscript is not well-written. My detailed questions are listed as follows.

A list of tools/papers can infer trait-relevant tissues:

1. Shang L, Smith J A, Zhou X. Leveraging gene co-expression patterns to infer trait-relevant tissues in genome-wide association studies[J]. PLoS genetics, 2020, 16(4): e1008734.
2. Calderon D, Bhaskar A, Knowles DA, Golan D, Raj T, Fu AQ, et al. Inferring Relevant Cell Types for Complex Traits by Using Single-Cell Gene Expression. Am J Hum Genet. 2017;101(5):686–99. WOS:000414251600003. pmid:29106824
3. Finucane, H. K., Reshef, Y. A., Anttila, V., Slowikowski, K., Gusev, A., Byrnes, A., et al. (2018). Heritability enrichment of specifically expressed genes identifies disease-relevant tissues and cell types. Nat. Genet. 50, 621–629. doi: 10.1038/s41588-018-0081-4
4. Pei, G., Dai, Y., Zhao, Z., and Jia, P. (2019). deTS: tissue-specific enrichment analysis to decode tissue specificity. Bioinformatics 35, 3842–3845. doi: 10.1093/bioinformatics/btz138

Authors' response: We thank the reviewer for these comments. Several points of concern are raised in this part, such as the comparison with other methods that we respond to below (in another “author’s response”). First, we want to clarify the integration of TWAS with LVs. As the reviewer says, TWAS results are one of our inputs. However, contrary to what the reviewer says, we are not comparing PhenoPLIER with TWAS as equivalent methods. We say that, when integrated with TWAS, LVs contain information that is not captured by TWAS, as shown in the LV246 example (aligned with lipid metabolism). TWAS associations allowed us to find that LV246 is strongly (and almost specifically) associated with lipids-related traits in both our discovery (Supplementary Table 8) and replication cohorts (Supplementary Table 9). However, key players such as

DGAT2 and *ACACA* (found in our CRISPR screen) were not associated with relevant traits in TWAS, and thus they were not contributing to LV-trait associations since their p-values are large. This suggests that LV246 contains more information that TWAS alone does not capture, but we are not saying that PhenoPLIER is "better" than TWAS. In fact, we provide an interpretation of these results based on the omnigenic model of complex traits, where TWAS hits might represent "peripheral" genes and some top-weighted LV members might be "core" genes.

Second, based on the *DGAT2/ACACA* example and LV246, the reviewer suggests that it would be easy to find functional trait-related genes using simple approaches that combine protein-protein interaction networks and pathways. We believe the reviewer is probably misunderstanding what we are doing in our work. The LVs used were extracted using an unsupervised learning method (PLIER), and some of them aligned well with known pathways, but others did not. An example of the former, which is the one selected by the reviewer, is LV246, aligned with lipid metabolism pathways (Supplementary Table 14) and associated with relevant traits, as we said before. On the other hand, our approach also found significant associations with LVs that do not align with prior knowledge but still represent biologically plausible and potentially novel disease-relevant mechanisms. An example is LV21 (Supplementary Figure 26 and Table 34), expressed in monocytes/macrophages (including microglia from cortex samples) and strongly associated with lipids and high cholesterol in the discovery and replication cohorts (Supplementary Figure 35 and Table 36). LV21 was also strongly associated with Alzheimer's disease in the discovery cohort (we note this disease is not present in the replication cohort). As we note in the Results section, macrophages play a key role in the reverse cholesterol transport and thus atherogenesis, and lipid metabolism in microglia has been recently identified as an important factor in the development of neurodegenerative diseases. Contrary to the reviewer's comment, these associations would not have been found using only prior knowledge (protein-protein interaction networks and known pathways).

Third, regarding the important concerns raised by the reviewer about potential false positives, we now included extensive simulations for our LV-based regression framework (Supplementary Note 1) across a thousand randomly-generated traits using data from 1000 Genomes. We ran the standard procedures of TWAS, which includes GWAS harmonization/imputation, and computing the gene-level statistics. We found acceptable mean type I error rates and approximately well-calibrated p-values under the null hypothesis of no associations. We also added simulations for our clustering analyses (Supplementary Note 2), which allowed us to verify that the clusters learned from the data significantly deviate from a scenario of no structure in the data.

We also made several changes across the manuscript to improve its readability, as the reviewer mentions here.

Besides, this method is based on multiple existing methods. My concern about this multi-step approach is the error accumulation. Hence, a negative control is necessary.

Authors' response: Yes, this is an important point that was added in this revision, as we mentioned before, with the simulations for our LV-based regression framework (Supplementary Note 1) and cluster analyses (Supplementary Note 2).

Comments:

· Does the method and its results account for false positives? the results lack a "negative control." Is there a way to perform simulation to assess this?

Authors' response: Yes, we have now added two Notes in the Supplementary Material regarding these negative controls:

- We added null simulations for our LV-based regression framework (Supplementary Note 1), which allowed us to verify that the model is approximately well-calibrated, with acceptable mean type I errors.
- We added simulations for our clustering analyses (Supplementary Note 2), which allowed us to verify that the structure learned from the data significantly deviates from a scenario of no structure.

· There does not appear to be any discussion of existing literature on competing methods. Using gene co-expression patterns to infer trait-relevant tissues is not novel in the field, such as CoCoNet. There are also many tools/papers (such as LDSC-SEG, deTS and RolyPoly) using cell/tissue specific expressed genes to infer trait-relevant tissues. Please compare PhenoPLIER results with the results of these tools on the same dataset (eg. 4,000 gene-trait associations used in this study).

Authors' response: We thank the reviewer for the comments on potentially related methods, such as those that infer tissue-specific effects. This comment made us revise the manuscript's introduction and improve how we present our approach. We acknowledge that the text excessively focused on inferring tissue-specific effects, which, although an important advantage of PhenoPLIER, is not the main focus. PhenoPLIER is a method that integrates genetic studies with gene modules extracted from expression data, and the ability to read the context where gene modules/LVs are expressed is important but optional. This is very clear from our results, where we could run all of our analyses without using the metadata in recount2, including the LV-based regression framework, drug-disease prediction, and cluster analyses.

We strongly believe that there are many fundamental differences between PhenoPLIER and the methods listed by the reviewer, where the only point that seemingly connects them is the tissue-specificity of effects. We feel that this reviewer's comment was on target given the previous framing of the manuscript, but with our more precise framing feel it has become less relevant. First, none of our main analyses can be performed with these other methods. Second, these other methods focus on producing a tissue ranking (or p-value) for a trait, whereas PhenoPLIER is about integrating gene modules with GWAS/TWAS. The context-specificity of gene modules/LVs is important to improve the interpretation of results, but it is entirely optional: we could perfectly go without this across all of our analyses. Third, the dataset used by the other methods to infer these tissue-specific effects is generally GTEx, a homogeneous expression dataset with a small number of samples compared with recount2 (the one used in MultiPLIER models). This is understandable because all these methods need a fixed and known list of tissues, such as CoCoNet, where a single tissue is processed at a time to produce a ranking. In contrast, in recount2, we do not know how many different tissues or contexts we have and producing this list would be highly challenging: For instance, the metadata in recount2 is incomplete or ambiguous between conditions; on the other hand, LVs/modules are expressed across several contexts (such as LV136, expressed in a wide range of cell types, including fibroblasts, mesenchymal stem cells, osteoblasts, pancreatic stellate cells, and cardiomyocytes). As we show in our results, although some LVs are clearly expressed in a single tissue, most of them are expressed across several contexts (sometimes samples with different diseases or under stimuli). Therefore, we believe that comparing PhenoPLIER with these methods is not only meaningless but also extremely hard to do, probably impossible, because the methods are fundamentally different. For the reasons above, using recount2 with these methods would be unfeasible.

Alternatively, in another attempt to follow the reviewer's suggestion to compare our approach with these other methods, we could apply PhenoPLIER using GTEx instead of recount2. This would imply running PLIER on GTEx first to obtain a gene latent representation, which could make the comparisons feasible. However, this new, GTEx-based representation would certainly extract a much smaller number of LVs, representing a downgrade, since GTEx is significantly less heterogeneous and large than recount2. But more importantly, even these LVs based on GTEx would very likely be expressed across different tissues. This, again, would make the comparison of PhenoPLIER with these other methods extremely blurry and hard to interpret. The

conclusion, as we previously said, is that PhenoPLIER and these other methods are fundamentally different approaches with different goals.

· The methodology seems incremental. Based on Figure 1b, PhenoPLIER, the method that the authors propose, simply multiplies Z transpose with M, which are outputs from two previously existing methods, MultiPLIER and PhenomeXcan (or LINCS), respectively.

Authors' response: Figure 1 was unclear about this, and we have now updated it to better reflect our work. PhenoPLIER consists of three main computational approaches: 1) an LV-based regression framework that computes a p-value between an LV and a trait, 2) cluster analyses of traits on the LV space using a consensus clustering approach, and 3) an LV-based drug-repurposing approach to predict drug-disease links. Multiplying two matrices is only a tiny aspect of the entire approach. The components of PhenoPLIER are now better explained in the introduction and the first section of Results (including the new Figure 1). We have also extensively updated the Methods section with more details about the methods developed and used.

· Why was a decision tree classifier (CART) chosen instead of other tree-based methods such as random forest? The latter usually has better performance and generalizability.

Authors' response: We use the classifier to aid interpretability, and the primary goal is not prediction performance. We specifically sought to understand which features were most discriminative for each cluster, which CART naturally provides. If we instead sought to build a predictive classifier, we agree that random forest would likely provide better generalizability. We have updated the Methods section to clarify that the goal was interpretability.

· Precision-recall curve decreases to 0.75 seems strange, is the scale correct?

Authors' response: In the gold standard set, 75% of drug-disease pairs are positive cases. In this scenario, a random classifier would have a precision of 0.75 by chance.

· In Figure 1a, the TWAS label should be near the arrows point from genes (middle) to the diseases (top). From the genotype to transcripts would be eQTL. Also in Figure 1a, why are the associations represented by directed arrows? The associations should not be undirected.

· Figure 1b: the dimension of Z transpose, which is latent variable (LV) by gene, is not properly represented in the figure. It should be a wide (rather than narrow) matrix. Also, at the top of the plot, the \sim or $\tilde{\sim}$ symbol should be an \approx symbol instead (based on my understanding).

Authors' response: We thank the reviewer for these detailed comments on our figures. We changed Figure 1 as suggested, and it now represents our proposal and its different components much better. We added a label explaining eQTLs and trait-associated variants from GWAS.

· The method description is vague. The correspondence between specific subsections of the results section and the methods section is unclear. The section "LV-trait associations via gene-property analysis" is hardly comprehensible.

Authors' response: Yes, we agree that the description of our methodology needed to be improved. We have significantly extended our descriptions in Methods, particularly the LV-based regression framework, which was updated after the reviewers' comments on the null simulations. We also reorganized the sections. The first subsection of Results ("PhenoPLIER: an integration framework based on gene co-expression patterns"), along with the updated Figure 1, now better explains the methodology developed in our study.

REVIEWER COMMENTS

Reviewer #1 (Remarks to the Author):

I thank the authors for responding to all my previous critique. I find the revised manuscript improved and largely addressing my previous comments. I maintain some of the critiques regarding my moderate enthusiasm; that being said I support the publication of the manuscript.

Reviewer #2 (Remarks to the Author):

The authors addressed my major concerns. Fig 1 is still challenging for readers to understand in a short time. In Fig 1a, I suppose the authors may highlight the inverse relationship between disease module and drug response module. Fig 1b, the logic is confusing, since it has information from both top to bottom and bottom to top. Lastly, the drug and traits cluster is not highlighted in a circle.

Reviewer #3 (Remarks to the Author):

I still cannot be convinced by the authors.

(1) Garbage in, garbage out! Regarding the false positive problem, the authors did not consider any false positive genes from the input TWAS results. It's well known that correlated expression and shared GWAS variants across individuals may cause false TWAS hits (Opportunities and challenges for transcriptome-wide association studies[J]. Nature genetics, 2019). The authors directly downloaded the TWAS results from PhenomeXcan and did not deal with these false TWAS hits.

For example, 1p13 locus for LDL, TWAS identified 34 genes for this locus, but only four genes may be the true target gene (Fig.6 from PMID 30926970; see the attached). It is impossible that PhenoPLIER can find meaningful results by taking so many false positive genes as input.

(2) I do not believe the rationale of PhenoPLIER, and the authors cannot provide evidence to show that expanding TWAS genes with a co-expression network does not include false positive genes.

(3) The motivation and novelty of this paper are not clear. As the authors said, the original manuscript “focused on inferring tissue-specific effects.” I previously questioned that there are already many tools that can infer tissue-specific effects. Accordingly, the authors changed their previous motivation to “However, it is hard to disentangle causal cell types using eQTL data alone, and other methods generally do not use the large amounts of publicly available RNA-seq data.” in the abstract to avoid comparing similar motivation tools. But the new motivation of PhenoPLIER is still not convincing to me.

[REDACTED]

Reviewer #4 (Remarks to the Author):

Please see attached document

This is an interesting piece of work offering a computational method that maps gene-trait associations and pharmacological perturbation data. Additionally, the authors have performed a CRISPR-Cas9 genome-wide loss-of-function screen to uncover and analyse lipid regulators, and to compare the results with MultiPLIER data. I have evaluated the technical accuracy of the CRISPR screen used in this revised manuscript, under review at Nature Communications. Since the other areas of expertise relevant to this manuscript are already covered by other reviewers, I will focus my comments on the CRISPR screening. The description of the CRISPR-screen related methods is accurate and gives enough detail about the experiments performed. There are some specifics that could be improved or corrected, detailed below;

Pag 25. Cell culture. It is not mentioned if the HepG2 cells were cultured in Collagen-I coated plates. HepG2 cells are usually cultured in Collagen-coated plates. This has been mentioned at the 'Lentiviral titer determination' section, so it is probably missing from 'cell culture' section.

Pag. 25. Lentiviral titer determination. The authors should review this sentence since it might mislead some readers; *"At MOI equal to or close to 0.3, around 95% of infected cells are predicted to have only one copy of the virus."*

The addition of this sentence may help the readers: *"At MOI of 0.3, ~25% of cells are infected, and 23% of cells are predicted to have only one copy of the virus (~95%)."*

Pag. 43-44. Figure S1-A. At the legend for panel A, it is not clear enough if the dots correspond to LipidSpotTM 488 or GFP signal. More detail would be helpful in this legend in general, but the dots labelling must be mentioned.

Pag. 44. Figure S2. If the samples shown in this figure were stained with LipidSpotTM 488, it would be helpful to add to this figure two details;

- i. The FACS analysis or dot-plot of the sample 'HepG2_lentiV2_Ctrl with no-viral transduction' stained with LipidSpotTM 488, to show that no signal is present in non-transduced cells.
- ii. Add that samples have been stained (or unstained) with LipidSpotTM 488.

Pag. 44-45. Figure S3-A and S3-B. Please add the type of DNA ladder used and the lane in which it has been run (i.e. Lane 0: 1kb DNA ladder) for both A and B panels.

Response to Reviewers' comments

We are most grateful for the opportunity to address the reviewers' additional comments and submit a further revised manuscript. A summary of the reviewers' comments and our response to each point made is presented below.

Reviewer #1

I thank the authors for responding to all my previous critique. I find the revised manuscript improved and largely addressing my previous comments. I maintain some of the critiques regarding my moderate enthusiasm; that being said I support the publication of the manuscript.

Authors' Response: We thank the reviewer for the insightful comments in the first revision that significantly improved our manuscript.

Reviewer #2

The authors addressed my major concerns. Fig 1 is still challenging for readers to understand in a short time. In Fig 1a, I suppose the authors may highlight the inverse relationship between disease module and drug response module. Fig 1b, the logic is confusing, since it has information from both top to bottom and bottom to top. Lastly, the drug and traits cluster is not highlighted in a circle.

Authors' Response: We thank the reviewer for the initial comments that allowed us to think critically about our assumptions in the PhenoPLIER framework and improve our manuscript.

We agree that Figure 1 continued to have opportunities for improvement. We made several adjustments to Figure 1 to incorporate the reviewer's suggestions and improve clarity:

Fig 1a:

- We made adjustments in this subfigure to improve clarity.
- We extended the caption of Fig 1a to incorporate the reviewer's suggestion about drugs and their effect on genes' expression: "[...] at the bottom-right corner, we show how a drug downregulates two genes (A and C)"

Fig 1b:

- Following the reviewer's comment, we made several adjustments in this subfigure to improve how we communicate the logic by making the flow of information go from top to bottom. At the top, we show the MultiPLIER models (matrix factorization of gene expression data) followed by the main input data types (TWAS and drug-induced transcriptional profiles from LINCS L1000). We moved the three computational components of PhenoPLIER to the bottom.
- Since we perform clustering of traits (but not drugs), we only highlighted a cluster of traits in this figure. The star-shaped points in the figure convey that this data type (drug-induced transcriptional profiles), as gene-trait associations, is projected into the same latent space for a joint analysis. We believe that the level of detail in the figure is enough to convey this intuition, whereas the Methods section provides all the details about the approach.
- We improved the picture of the "LV-based regression model" to more accurately reflect the model we describe in the Methods section. We added the multivariate normal distribution of the error terms to show more directly that the model accounts for gene-gene correlations.

We also extended and improved the caption of Figure 1 to more clearly explain all the parts and their connections.

Reviewer #3

I still cannot be convinced by the authors.

(1) Garbage in, garbage out! Regarding the false positive problem, the authors did not consider any false positive genes from the input TWAS results. It's well known that correlated expression and shared GWAS variants across individuals may cause false TWAS hits (Opportunities and challenges for transcriptome-wide association studies[J]. Nature genetics, 2019). The authors directly downloaded the TWAS results from PhenomeXcan and did not deal with these false TWAS hits.

For example, 1p13 locus for LDL, TWAS identified 34 genes for this locus, but only four genes may be the true target gene (Fig.6 from PMID 30926970; see the attached). It is impossible that PhenoPLIER can find meaningful results by taking so many false positive genes as input.

(2) I do not believe the rationale of PhenoPLIER, and the authors cannot provide evidence to show that expanding TWAS genes with a co-expression network does not include false positive genes.

Authors' Response: Both points (1) and (2) above relate to potential false positives. We took the concerns that all reviewers expressed about potential false positives seriously, and their feedback substantively improved the manuscript. We addressed these by improving our regression model and conducting extensive simulations that allowed us to verify that p -values are calibrated (Supplementary Note 1).

We agree with the reviewer that correlated expression can cause false TWAS hits. This is why we explicitly model gene-gene correlations in TWAS statistics. The statement that we did not address false TWAS hits arising from this phenomenon is incorrect. We show that the model is well-calibrated for gene modules (or latent variables) with highly correlated genes at the top (in our previous revision, these are Supplementary Figures 9, 10, 11, and 12). For example, 22 genes from the top 70 genes (1%) of LV234 are from region 6p22 and encode different histones. The high correlation between these genes makes a standard OLS model fail, whereas our approach yields well-calibrated p -values (Supplementary Figure 9). We provide several examples of gene modules where our model correctly accounts for correlations.

The reviewer says it is “*well known that correlated expression and shared GWAS variants across individuals may cause false TWAS hits,*” citing the same article (Wainberg, 2019) we also cited in our manuscript. For instance, in the Methods section (under “Regression model for LV-trait associations”), where we fully define our regression model and how we compute the gene-gene correlations, we say:

Since the error terms ϵ could be correlated, we cannot assume they have independent normal distributions as in a standard linear regression model. In the PrediXcan family of methods, the predicted expression of a pair of genes could be correlated if they share eQTLs or if these are in LD (Wainberg, 2019). Therefore, we used a generalized least squares approach to account for these correlations. The gene-gene correlation matrix R was approximated by computing the correlations between the model sum of squares (SSM) for each pair of genes under the null hypothesis of no association. [...]

In the Discussion section, we say:

[...] Additionally, TWAS has several limitations that can lead to false positives (Wainberg, 2019; Hukku, 2021). Like GWAS, which generally detects groups of associated variants in linkage disequilibrium (LD), TWAS usually identifies several genes within the same locus (Gamazon, 2015; Gusev, 2018). This is due to sharing of GWAS variants in gene expression models, correlated expression of nearby genes, or even correlation of their predicted expression due to eQTLs in LD, among others (Wainberg, 2019). Our LV-based regression framework, however, accounts for these gene-gene correlations in TWAS reasonably well.

Reviewer #3's assertion that we "directly downloaded the TWAS results from PhenomeXcan and did not deal with these false TWAS hits" does not account for the significant work that we did in the revised manuscript directly addressing this concern. To further clarify that we account for potential false positives from gene correlations, we updated Figure 1b in this new revision to directly show that the error terms in the regression framework follow a multivariate normal distribution using the gene-gene correlation matrix.

When the reviewer says that "For example, 1p13 locus for LDL, TWAS identified 34 genes for this locus, but only four genes may be the true target gene [...]", they seem concerned about our results with a gene module associated with lipid metabolism (LV246, Results section, under "LVs link genes that alter lipid accumulation with relevant traits and tissues"). We did not find 1p13 driving LV246, as none of the top genes in LV246 were from that region. The reviewer's concern is exactly why we correct for highly correlated genes.

The reviewer also says that "It is impossible that PhenoPLIER can find meaningful results by taking so many false positive genes as input." First, PhenoPLIER findings were replicated in an independent cohort (eMERGE). We also found that PhenoPLIER 1) LV-trait associations are biologically meaningful (such as LV246, with genes expressed in adipose tissue, and statistically significant associations with highly lipid-related traits in both our discovery and replication cohorts), 2) detects meaningful and expected clusters of traits in the LV space (Figures 6 and 7), and 3) when integrated with another data type, such as transcriptional responses to small molecule perturbations, the LV-based drug repurposing approach outperformed a previously published single-gene approach (Figure 3). Taken together, our simulations and these results strongly support a model where PhenoPLIER does find meaningful results and are incompatible with a model where results are driven simply by the noise of false positives.

We designed PhenoPLIER knowing that TWAS results would include false positives. However, the regression model takes into account *all* genes in the LV (those with high and also low weights), as defined in Equation 7, where we say that the model tests "whether top-weighted genes for an LV are more strongly associated with the phenotype than other genes with relatively small or zero weights." Therefore, false positives in TWAS could also lead to false negatives in our LV-based regression model, which we acknowledge in the Methods section. Furthermore, as mentioned in the Methods section, our regression model is based on the established gene-set analysis framework in MAGMA (de Leeuw, 2015), which we adapted for TWAS (specifically, for S-MultiXcan).

In summary, we agree with the reviewer that false positives from correlated expression could confound TWAS-based approaches. However, the reviewer's important points on the original submission led us to directly examine this. We observed the effect described and adjusted our regression model to directly address it. The reviewer's feedback on the initial version substantially strengthened this work and will likely improve its utility and generalizability.

(3) The motivation and novelty of this paper are not clear. As the authors said, the original manuscript "focused on inferring tissue-specific effects." I previously questioned that there are already many tools that can infer tissue-specific effects. Accordingly, the authors changed their previous motivation to "However, it is hard to disentangle causal cell types using eQTL data alone, and other methods generally do not use the large

amounts of publicly available RNA-seq data.” in the abstract to avoid comparing similar motivation tools. But the new motivation of PhenoPLIER is still not convincing to me.

Authors' Response: The reviewer says that “*the authors changed their previous motivation [...] in the abstract to avoid comparing similar motivation tools.*” We agreed with the reviewer’s previous point that our earlier motivating statement was imprecise. We feel that the change we made in response to the reviewer’s feedback strengthened the work. In our previous responses regarding this point, we noted that the tools cited by the reviewer and PhenoPLIER were “*fundamentally different approaches with different goals.*” We can run all of our main analyses without reading the cell type/tissue where gene modules/LVs are expressed. The cited methods: 1) cannot perform any of our main analyses; 2) their main goal is to generate a tissue-trait ranking/p-value that is entirely optional in PhenoPLIER; 3) they necessarily use relatively homogeneous datasets with a previously known/fixed list of tissues, whereas we use a highly heterogeneous dataset where it is very hard, likely impossible, to compute a summary of different tissues or cell types. As with the previous points, the reviewer’s initial comments led to a significant improvement and refinement in the manuscript by encouraging us to make our framing precise. We hope that the editors and reviewers would agree that taking into account the reviewer’s feedback strengthened the manuscript.

Reviewer #4

This is an interesting piece of work offering a computational method that maps gene-trait associations and pharmacological perturbation data. Additionally, the authors have performed a CRISPR-Cas9 genome-wide loss-of-function screen to uncover and analyse lipid regulators, and to compare the results with MultiPLIER data. I have evaluated the technical accuracy of the CRISPR screen used in this revised manuscript, under review at Nature Communications. Since the other areas of expertise relevant to this manuscript are already covered by other reviewers, I will focus my comments on the CRISPR screening. The description of the CRISPR-screen related methods is accurate and gives enough detail about the experiments performed. There are some specifics that could be improved or corrected, detailed below;

Authors' Response: We thank the reviewer for the very positive feedback and the useful comments on our CRISPR screen that we address below.

Pag 25. Cell culture. It is not mentioned if the HepG2 cells were cultured in Collagen-I coated plates. HepG2 cells are usually cultured in Collagen-coated plates. This has been mentioned at the ‘Lentiviral titer determination’ section, so it is probably missing from ‘cell culture’ section.

Authors' Response: We thank the reviewer for pointing this out. HepG2 cells were maintained and cultured in Collagen-coated flasks. We have added this information to the “Cell Culture” paragraph: “*Cells were kept at 37oC in a humidity-controlled incubator with 5% CO2, and were maintained at a density not exceeding more than 80% confluency in Collagen-I coated flasks.*”

Pag. 25. Lentiviral titer determination. The authors should review this sentence since it might mislead some readers; “At MOI equal to or close to 0.3, around 95% of infected cells are predicted to have only one copy of the virus.”

The addition of this sentence may help the readers: “At MOI of 0.3, ~25% of cells are infected, and 23% of cells are predicted to have only one copy of the virus (~95%).”

Authors' Response: We are grateful for the reviewer's important clarification. We have corrected the sentences to avoid this confusion as follows: *"At MOI close to 0.3, around 25% of cells are infected, and the majority of those infected cells are predicted to have only one copy of the virus."*

Pag. 43-44. Figure S1-A. At the legend for panel A, it is not clear enough if the dots correspond to LipidSpot™ 488 or GFP signal. More detail would be helpful in this legend in general, but the dots labelling must be mentioned.

Authors' Response: Thank you. Details have been added in the legend: *"Figure S1: **EVOS Fluorescence Microscope Image Capture.** A. HepG2_lentiV2_Ctrl with no-viral transduction. B. HepG2_lentiV2 with viral transduction. Both no-viral transduction Control (A) and lentiviral transduction (B) HepG2 cells were stained with LipidSpot™ 488."*

Pag. 44. Figure S2. If the samples shown in this figure were stained with LipidSpot™ 488, it would be helpful to add to this figure two details;

- i. The FACS analysis or dot-plot of the sample 'HepG2_lentiV2_Ctrl with no-viral transduction' stained with LipidSpot™ 488, to show that no signal is present in non-transduced cells.
- ii. Add that samples have been stained (or unstained) with LipidSpot™ 488.

Authors' Response: Thank you. Corrected as follows in the legend: *"Figure S2: **Fluorescence-Activated Cell Sorting Gate Setting.** A. HepG2_UnStained WT. B. HepG2_lentiV2 with viral transduction stained with LipidSpot™ 488. HepG2_lentiV2 cells were FAC sorted, 20% of GFP-High and 20% of GFP-Low cell populations were collected."*

Pag. 44-45. Figure S3-A and S3-B. Please add the type of DNA ladder used and the lane in which it has been run (i.e. Lane 0: 1kb DNA ladder) for both A and B panels.

Authors' Response: Thank you. The type of ladder was added at the end of the legend: *"For both panels A and B, a 100 bp ladder was used in Lane 0."*

REVIEWERS' COMMENTS

Reviewer #2 (Remarks to the Author):

As I mentioned in my previous comments. The authors have addressed most of my critiques.

Reviewer #3 (Remarks to the Author):

The authors significantly improved their analyses. I still have one comment about how the authors addressed the false positives.

The authors somewhat address the false positives by comparing the type I errors in simulation. Specifically, they compare the QQ plots for each of the LVs and report that Suppl Fig 9 - 12 shows calibrated type I errors for 4 chosen LVs out of 987. But Suppl Figure 8 seems to show that p-values are not as well-calibrated under the GLS model as claimed. I wonder if the authors would consider adding a box plot to show the mean type I error for every 987 LVs between OLS and GLS models? Or at least justify why they selected these 4 LVs (corresponding to Suppl Fig 9 - 12).

Response to Reviewers' Comments

We appreciate the opportunity to respond to the reviewers' additional comments and submit an updated version of our manuscript. Below, you will find a summary of the reviewers' feedback and our corresponding responses to each point.

Reviewer #2

As I mentioned in my previous comments. The authors have addressed most of my critiques.

Authors' Response: We thank the reviewer for the insightful comments in the previous revisions that significantly improved our manuscript.

Reviewer #3

The authors significantly improved their analyses. I still have one comment about how the authors addressed the false positives.

The authors somewhat address the false positives by comparing the type I errors in simulation. Specifically, they compare the QQ plots for each of the LVs and report that Suppl Fig 9 - 12 shows calibrated type I errors for 4 chosen LVs out of 987. But Suppl Figure 8 seems to show that p-values are not as well-calibrated under the GLS model as claimed. I wonder if the authors would consider adding a box plot to show the mean type I error for every 987 LVs between OLS and GLS models? Or at least justify why they selected these 4 LVs (corresponding to Suppl Fig 9 - 12).

Authors' Response: We thank the reviewer for this additional comment on false positives, which we take very seriously in our work and previous responses.

The reviewer is correct in noting the slight deviation from the expected uniform distribution in the QQ plots, and we acknowledged this limitation of the approach in the Methods and Discussion sections of our manuscript. We selected these 4 LVs (and also the one shown in Suppl. Figure 13) as part of our attempt to understand this deviation. In this analysis, we found two groups of LVs:

- 1) The first group consists of LVs with many adjacent genes at the top, which are expected to have highly correlated TWAS statistics (because of shared eQTLs, etc., as mentioned in the Methods section). Some examples of these LVs are shown in Suppl. Figures 9 - 12 (those mentioned by the reviewer), where the baseline OLS model produces higher mean type I errors as expected since it does not account for correlated TWAS statistics. Our GLS model, however, accounts for these correlations using matrix \mathbf{R} (gene-gene correlation matrix) and thus produces well-calibrated p-values for these LVs.
- 2) The second group consists of LVs with higher-than-expected mean type I errors in both the GLS and OLS models, although they don't have a relatively large number of adjacent genes at the top. We show one example in Suppl. Figure 13 (LV914). As we say in Supplementary Note 1, *"Inflation in these LVs might be explained by inaccuracies in correlation estimates between the individual-level MultiXcan model and its summary-based version."*

The Methods section explains this in more detail, where we say:

Note that, since we used the MultiXcan regression model (Equation (3)), \mathbf{R} is only an approximation of gene correlations in S-MultiXcan. As explained before, S-MultiXcan

approximates the joint regression parameters in MultiXcan using the marginal regression estimates from S-PrediXcan in (2) with some simplifying assumptions and different genotype covariance matrices. This complicates the derivation of an S-MultiXcan-specific solution to compute \mathbf{R} . To account for this, we used a submatrix \mathbf{R}_ℓ corresponding to genes that are part of LV ℓ only (top 1% of genes) instead of the entire matrix \mathbf{R} . This simplification is conservative since correlations are accounted for top genes only. [...]

Our action regarding the LVs in the second group was to remove them from our analyses, as we previously noted in Supplementary Note 1: *“Therefore, we flagged those with a type I error rate larger than 0.07 (127 LVs) and excluded them from our main analyses.”*

Based on the reviewer’s feedback, we agree that this is helpful to have in a more central location in the manuscript, so we have added this information to the previous paragraph of the Methods section: *“Our simulations (Supplementary Note 1) show that the model is approximately well-calibrated and can correct for LVs with adjacent and highly correlated genes at the top (e.g., Figure 9). The simulation also identified 127 LVs in which the model was not well-calibrated (e.g., Figure 13). As explained before, this can be attributed to the limitations in accurately computing a gene correlation matrix and as a result, we have excluded these LVs from our main analyses.”*